# Parallelizing analog in-sensor visual processing with arrays of gate-tunable silicon photodetectors

Zheshun Xiong[1,3], Wen Liang[1,3], Meiyue Zhang[1,3], Dacheng Mao[1], Qiangfei Xia [1] & Guangyu Xu [1,2] ✉

In-sensor processing of dynamic and static information of visual objects avoids exchanging redundant data between physically separated sensing and computing units, holding promise for computer vision hardware. To this end, gate-tunable photodetectors, if built in a highly scalable array form, would lend themselves to large-scale in-sensor visual processing because of their potential in volume production and hence, parallel operation. Here we present two scalable in-sensor visual processing arrays based on dual-gate silicon photodiodes, enabling parallelized event sensing and edge detection, respectively. Both arrays are built in CMOS compatible processes and operated with zero static power. Furthermore, their bipolar analog output captures the amplitude of event-driven light changes and the spatial convolution of optical power densities at the device level, a feature that helps boost their performance in classifying dynamic motions and static images. Capable of processing both temporal and spatial visual information, these retinomorphic arrays suggest a path towards large-scale in-sensor visual processing systems for high-throughput computer vision.

Computer vision has become an increasingly significant technology in autonomous navigation[1], object recognition[2], bioimaging[3,4], and human-machine interfacing[5]. Yet, the ubiquity of time-sensitive and data-intensive computer vision tasks brings a growing challenge for existing vision systems, which often involve exchanging redundant data between physically separated sensing and computing units. To this end, in-sensor processing of dynamic and static visual information has risen to be a viable hardware approach to lessen the latency and energy consumption spent over the data exchange by integrating sensing and pre-processing units at the device level[6,7]. Such in-sensor processing hardware emulates the way the human retina[8–11] acts to trace temporal changes[1,12–16](e.g., motion) and extract spatial features[17–21] (e.g., edges), drawing broad interest in associated circuit designs[22–25], analytical algorithms[26,27], and intelligent systems[2,13,18–20,27,28].

Among emerging in-sensor visual processors, gate-tunable photodetectors based on silicon[29], ferroelectrics[30], and nanomaterials[1,14,18,20,21,31] pose a solution to form scalable processor arrays for high parallelism. These devices take their gate-tunable photoresponsivity ($R_{ph}$) as the weight for in situ multiply-accumulate operation[27,29], cutting the computational overhead in circuits[12,15–17], graphics processing units (GPU)[32], or field-programmable gate arrays (FPGA)[22] that are commonly adopted in visual processing systems. Moreover, their optical responses in the analog domain hold promise to capture temporal and spatial visual information needed for visual processing. Nonetheless, most of these device prototypes have yet to be tested for both static and dynamic visual processing; their performance in recognizing sophisticated objects at an array level needs to be further examined[33]. For these reasons, it is imperative to develop ideally scalable, compact, and low-

[1]Department of Electrical and Computer Engineering, University of Massachusetts, Amherst, Massachusetts 01003, USA. [2]Department of Biomedical Engineering, University of Massachusetts, Amherst, Massachusetts 01003, USA. [3]These authors contributed equally: Zheshun Xiong, Wen Liang, Meiyue Zhang. ✉e-mail: guangyux@umass.edu

power arrays to detect dynamic events and extract static features with a high degree of parallelism. This is a non-trivial task as it requires a holistic modular-array co-design that takes key figure-of-merits (e.g., uniformity, crosstalk, power) into a well-balanced account.

Here, we present two scalable in-sensor visual processing arrays tailor-designed for parallelized event sensing and edge detection, respectively, based on dual-gate amorphous-silicon photodiodes (α-Si PDs) placed in *ca.* 200 μm pitches. Both arrays consume zero static power by short-circuit operations; their bipolar analog output directly quantifies event-driven light changes (i.e., temporal visual processing) and light intensities on the edge of light spots (i.e., spatial visual processing) at the device level, respectively. Such analog in-sensor computing capability empowers these arrays to process sophisticated visual objects with parallelism and well perform the classification tasks among dynamic motions and static images. Specifically, the first array parallelizes computing units (CU) that comprise integrated PDs, resistors (R), and capacitors (C), whose bipolar analog responses act to detect site-selective light pulses at sub-ms precision. We numerically show that these CUs can form ten 120-by-160 arrays to classify sophisticated human motions (with 90% accuracy) via an offline spike neural network (SNN). On the other hand, the second array parallelizes image kernels that comprise 3-by-3 PDs, whose bipolar analog responses act to simultaneously identify the edges of multiple objects by convolutional filtering. We numerically show that an array of 8-by-8 kernels can be reconfigured (via gating) to 10 filters to classify handwritten digits (with 94.8% accuracy) via an artificial neural network (ANN). Capable of processing both temporal and spatial visual information in the analog domain, our PD-based retinomorphic arrays suggest a path towards low-power, large-scale in-sensor visual processing systems for high-throughput computer vision.

## Results

### Modular design of gated Si photodiodes

Leveraging crystalline Si-based dual-gate PDs reported by Jang, H. et al[29]., our engineering efforts start from re-designing these gate-tunable p-i-n PDs as the analog visual processing unit with the following improvements (Supplementary Fig. 1).

First, the materials of the PDs are different. Instead of building PDs from an intrinsic crystalline silicon substrate, here we deposit intrinsic α-Si films on top of a $SiO_2$/Si wafer (sandwiched by oxide and metallization layers, see Methods) to form photo-sensitive regions of the diode. This change is made for the high absorption coefficient of α-Si (vs. crystalline Si) and its compatibility with monolithic integration onto complementary metal–oxide–semiconductor (CMOS) chips (vs. intrinsic Si substrate).

Second, the filling factor (FF) of PDs is increased. Different from prior PD structure[29], here we leave no metal contacts on top of the photo-sensitive α-Si regions. As a result, each PD is fully exposed to the incident light, yielding an increased FF.

Third, the device layout is more error-tolerant. Here we place an even number of α-Si channels between each pair of the source and drain contacts (S/D). This layout minimizes the dependence of the photoresponse on the possible asymmetric electrostatic doping effect caused by alignment error of gate contacts (G1/G2, see Supplementary Fig. 2), keeping a similar range of the absolute photoresponse values when the two gate biases flip their polarities at the same time.

Lastly, the PD dimension is scaled down to build compact visual processing arrays. We reduce the size of the active region in each PD down to *ca.* 70–80 μm, and the channel width/length ratio down to *ca.* 400–470/5 μm (vs. 300 μm and 5576/5 μm in ref. 29).

### Fabrication of individual Si photodiodes

With the foregoing design considerations, we first form gate-routing lines by sputtering Ti/Pt layers on top of a $SiO_2$/Si substrate (Methods) and passivate them with a 300 nm $SiO_2$ layer deposited by plasma-enhanced chemical vapor deposition (PECVD). Next, we form Cr/Au-based vias (10/300 nm) through this passivation layer (by via opening and metallization steps) and connect them with two interdigitated gate contacts in the shape of multi-fingers (G1 and G2, the finger pitch [finger width] is 15 μm [*ca.* 70–80 μm]) based on sputtered Ti/Pt layers. We then use an atomic layer deposition (ALD) step to form an $Al_2O_3$-based gate oxide layer (*ca.* 30 nm), followed by making S- and D-contacts on top with Ti/Pt layers. These S- and D-contacts are centered to G1- and G2-contacts, respectively, but chosen to be 2-μm narrower; this device geometry assures that G1 [G2] can create electrostatically doped regions surrounding S[D]-contacts (Fig. 1a). We then deposit a PECVD-based intrinsic α-Si film (*ca.* 250 nm) on top of the S- and D-contacts, and pattern it into the active region of the PD. The entire device is finally passivated by a PECVD-$SiO_2$ layer (*ca.* 300 nm), and routed to wire-bonding pads with Ti/Pt layers.

### Optoelectronic characteristics of Si photodiodes

Our dual-gate PD structure serves to alter *both* the direction *and* the amplitude of diode current by two independent gate biases ($V_{G1}$ and $V_{G2}$ on G1- and G2-contacts) via gate-induced electrostatic doping effect in the α-Si film. To assess such electrostatic doping effects[29,34–36] in our PDs, we set the two gate biases as $V_{G1} = -V_{G2} = 3$ or $-3$ V, and measured the source current $I_S$ in the dark when $V_S = -V_D$ was swept from -3 to 3 V at a step of 50 mV (Fig. 1b, c). In this configuration, the field effect from the positively [negatively] biased gate will create n-type [p-type] electrostatic doping profiles in the α-Si regions above them. The measured $I_S - V_S$ curves are rectified with a turn-on voltage at *ca.* 0.7 or $-0.7$ V, suggesting the existence of electrostatically doped p-i-n/n-i-p regions between S- and D-contacts (see simulation results in Supplementary Fig. 3). Next, under constant optical power density ($P_{light} = 530$ mW cm$^{-2}$ at 550/15 nm, an illumination centered at 550 nm with a 15 nm bandwidth), $I_S - V_S$ curves measured at $V_{G1} = V_{G2} = 3$ or $-3$ V feature higher $I_S$ values than those measured with floated G1 and G2 (Supplementary Fig. 4), possibly because the gate-induced p-i-p/n-i-n doping profile reduces the channel resistance[37]; these linear $I_S - V_S$ curves also suggest insignificant Schottky barriers near S- and D-contacts[35,36].

We next investigate gate-dependence of the $I_{ph}$ in our PDs (i.e., the $I_S$ under light illumination subtracted by that in the dark), an essential figure of merit for in-sensor visual processing[6,7,18]. To this end, we map short-circuited $I_{ph}$ values (i.e., measured at $V_S = V_D = 0$ V, $P_{light} = 35$ mW cm$^{-2}$ at 595 nm to exemplify the PD response in the visible spectrum) with $V_{G1}$ and $V_{G2}$ being swept from $-3$ to $+3$ V at a step of 200 mV, and identify four distinct operation regions on the map (Fig. 1d):

First, we observe n-i-n and p-i-p regions when $V_{G1}$ and $V_{G2}$ are nearly equal. These two regions show low $I_{ph}$ values because the built-in potentials across n-i [p-i] and i-n [i-p] junctions cancel each other. These regions are noted to deviate from the diagonal line, $V_{G1} = V_{G2}$, which can result from charge-trapping defects within α-Si films or near the surface of the gate-oxide layer[29] (requiring extra gate biases to dope the channel). The $I_{ph}$ values in the p-i-p region are lower than those in the n-i-n region, likely because the work function of S/D-contacts (Ti/Pt) is closer to the conduction band edge of α-Si[35,36].

Second, we observe n-i-p and p-i-n regions when $V_{G1}$ and $V_{G2}$ are nearly opposite to each other. In these two regions, $I_{ph}$ monotonically increases from a negative maximum to zero, and further increases to a positive maximum along the negative diagonal line, $V_{G1} = -V_{G2} = V_p$ from $-3$ to 3 V. This trend follows the sum of the built-in potential across n-i [p-i] and i-p [i-n] junctions, which decreases its amplitude to zero when $V_p$ changes from $-3$ to 0 V, then flip its sign and increase its amplitude when $V_{G1}$ changes from 0 to 3 V. Such gate-tunability of $I_{ph}$ (Fig. 1e) – in terms of its direction and amplitude – is the key feature of our PDs for analog visual processing; for the rest of this work, we always gate them as $V_{G1} = -V_{G2} = V_p$.

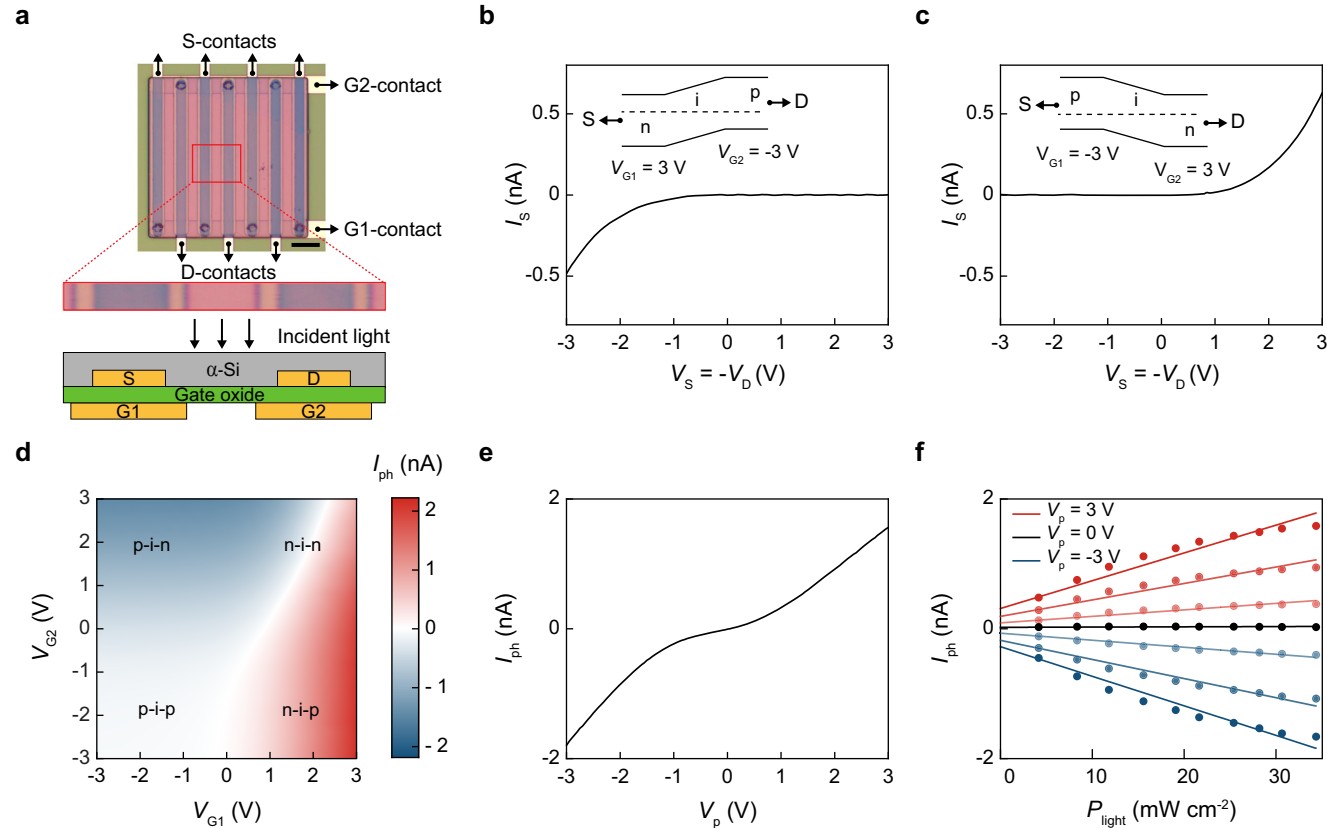

**Fig. 1 | Miniaturized dual-gate silicon photodetectors with gate-tunable photoresponse. a** A dual-gate α-Si PD with its zoom-in view and cross-sectional structure. Scale bar, 100 μm. **b** $I_S$ – $V_S$ curves measured in dark with $V_{G1} = -V_{G2} = 3$ V. **c** $I_S$ – $V_S$ curves measured in dark with $V_{G1} = -V_{G2} = -3$ V. In (**b**) and (**c**), the insets show their associated band diagrams at $V_S = V_D = 0$ V. **d** Contour plots of short-circuited $I_{ph}$ values measured with $V_{G1}$ and $V_{G2}$ ranging from $-3$ to $+3$ V, $P_{light} = 35$ mW cm$^{-2}$ centered at 595 nm. **e** The $I_{ph}$ - $V_p$ curve measured at $V_S = V_D = 0$ V, $P_{light} = 35$ mW cm$^{-2}$ centered at 595 nm. **f** $I_{ph}$ - $P_{light}$ curves measured at $V_S = V_D = 0$ V, with $V_p$ ranging from $-3$ to 3 V at a 1 V step, $P_{light} = 0$–35 mW cm$^{-2}$ centered at 595 nm.

Finally, we examine the linearity of our PDs biased at various $V_p$ values ($P_{light} = 0 – 35$ mW cm$^{-2}$ centered at 595 nm, Fig. 1f; see additional experiments in Supplementary Fig. 5). Our results show that $I_{ph}$ linearly increases with $P_{light}$ ($R^2 > 0.91$) for $P_{light}$ up to 25 mW cm$^{-2}$ and starts to saturate at higher $P_{light}$ due to the limited carrier lifetime of α-Si that may occur at a high density of photoinduced carriers[38]. When $V_p$ changes from $-3$ to 3 V, the slope of $I_{ph}$-$P_{light}$ curves increases from a negative maximum to a positive maximum, confirming the gate-tunability of $I_{ph}$ in terms of their polarities and amplitudes.

## Pairing photodiodes for event detection

Event-based vision sensors emulate the human retina to capture the temporal changes of $P_{light}$ ($\Delta P_{light}$, i.e., events) in the field of view (FOV), circumventing data-intensive inter-frame differentiation steps in standard CMOS imagers[16,39]. Among them, dynamic vision sensors and asynchronous time-based image sensors[12,16,40,41] are industry standards for event sensing, but often at the expense of hardware/design overhead on in-pixel/peripheral circuits. On the other hand, emerging optoelectronic synaptic devices[19,31,42,43] and 2D-material-based photodetectors[44,45] are viable alternatives for their structural simplicity and promise for in-sensor SNN, respectively; they, however, would benefit from further studies on their temporal precision, reliability, and feasibility for mass production of large-scale arrays. To this end, we leverage our compact, gate-tunable, and low-power α-Si PDs to build an integrated event-based vision processor, showcasing the capability of these PDs for analog processing of temporal visual information. To achieve this, we pair up two PDs and connect them with two Rs and one C in a monolithic manner to form a compact in-sensor CU (i.e., 2PD-2R-1C circuit computing unit, Fig. 2a); R and C are formed by a PECVD-based n-doped α-Si

layer (100 nm) and an ALD-based HfO$_2$ layer (15 nm) sandwiched between metal layers, respectively, and routed to PDs or testing pads (Methods and Supplementary Fig. 6).

From the circuit perspective, the two α-Si PDs placed in two parallel branches are gated as n-i-p and p-i-n diodes, respectively, leading to the same amount of $I_{ph}$ that flows to opposite directions (i.e., opposite signs of the photoresponse in Supplementary Fig. 7, see $V_p$ values in Supplementary Tab. 1); the photoresponse from the 1R1C branch is expected to respond to $\Delta P_{light}$ slower than that in the 1R branch due to the extra RC time delay. Moreover, our in-sensor CU consumes zero static power for visual processing[11], since both branches are short-circuited by grounding S-contacts of PDs and the input of a trans-impedance amplifier (TIA, Fig. 2b). The net current flows into a TIA – the difference of $I_{ph}$ in two branches (if any) – converting to a readout voltage $V_{out}$. Under this configuration, our CU outputs zero $V_{out}$ when $P_{light}$ is kept as a constant (i.e., no events); when $P_{light}$ changes, the 1R1C branch responds to $\Delta P_{light}$ with a latency compared to the 1R branch (exemplified by $R_1 = R_2 = 100$ MΩ, $C_1 = 100$ pF, Fig. 2c), resulting a positive [negative] spike (i.e., ON/OFF spike) when $P_{light}$ increases [decreases]. Such ON/OFF spikes consistently occur near the rising/falling edges of every light pulse we apply ($\Delta P_{light} = 530$ mW cm$^{-2}$ at 550/15 nm, $t_{on}/t_{off} = 90/130$ ms, three independent 20-pulse periods), showing reliable in-sensor event detection.

We next characterize the shape of these ON/OFF spikes by varying gate-tunable photoresponse and $\Delta P_{light}$, respectively (Fig. 2d–i, see $V_p$ values in Supplementary Tab. 1). With a constant $\Delta P_{light}$ (530 mW cm$^{-2}$ at 550/15 nm, $t_{on}/t_{off} = 90/130$ ms), spike amplitudes, $A_{on}$ and $A_{off}$ (defined as positive/ negative maximum subtracted by 10-point average in the baseline), are found to increase with $V_p$ ranging from 1.0 to 2.5 V (Fig. 2d, e); with a constant $V_p$, $|A_{on}|$ and $|A_{off}|$ are found to increase with

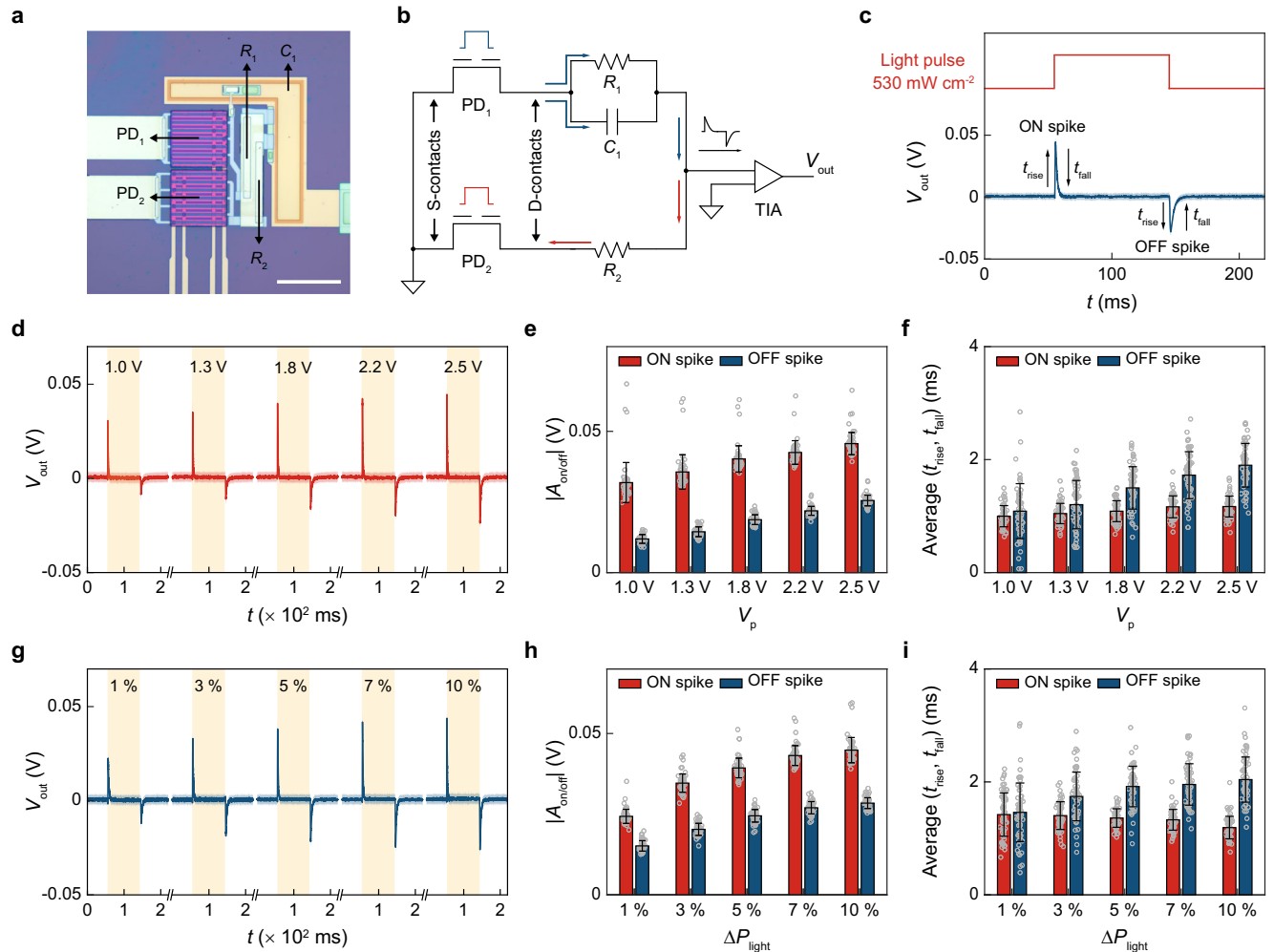

**Fig. 2 | Pairing dual-gate photodetectors for analog in-sensor event detection.** **a** Optical image of a single CU. Scale bar, 100 μm. **b** The equivalent circuit of a single CU. **c** CU response to 550/15 nm light pulses ($\Delta P_{light}$ = 530 mW cm⁻², $t_{on}/t_{off}$ = 90/130 ms, three 20-pulse periods). **d** $V_p$-dependence of the $V_{out}$ traces from a single CU; the light pulsing condition in (**c**) is repeated here. **e** $|A_{on}|$ and $|A_{off}|$ extracted from (**d**). **f** The average of $t_{rise}$ and $t_{fall}$ extracted from (**d**). **g** $\Delta P_{light}$-dependence of the $V_{out}$ traces from a single CU ($V_p$ is fixed at 2.5 V); $\Delta P_{light}$ ranges from 53 (1%) to 530 (10%) mW cm⁻², while the rest of the light pulsing condition is the same as (**c**). **h** $|A_{on}|$ and $|A_{off}|$ extracted from (**g**). **i** The average of $t_{rise}$ and $t_{fall}$ extracted from (**g**). Shaded areas in (**c**, **d**, and **g**) and error bars in (**e**, **f**, **h**, and **i**) both represent ±1 standard deviation (S.D.) from a total of 60 pulses in three light pulsing periods.

$\Delta P_{light}$ ranging from 53 to 530 mW cm⁻² (Fig. 2g, h). Such $V_p$- and $\Delta P_{light}$-dependent spike amplitudes demonstrate the capability of our CUs for analog in-sensor visual processing; the gate-tunable synapse-like behaviors can be used to form SNN[28,43–45] and further develop AI chips[46,47]. The mismatch between $|A_{on}|$ and $|A_{off}|$, while not affecting event sensing, can be effectively reduced by optimizing the experimental setting (Supplementary Fig. 8). On the other hand, the average of the rising and falling times of these spikes ($t_{rise}$ [$t_{fall}$] from 10 to 90% [90 to 10%] $|A_{on/off}|$ change) is mostly <3 ms across all $V_p$ and $\Delta P_{light}$ values (Fig. 2f, i, TIA configured to a high-bandwidth mode). This ms-level temporal precision is on par with the response speed of human retina[11] and suffices the requirement of latency-sensitive applications[1,4]. If we further reduce RC values (<100 MΩ/100 pF) and increase the bandwidth of TIA (>1 MHz), our CUs can ultimately respond to light pulses with < 2 μs $t_{rise}$ and < 11 μs $t_{fall}$ due to the small RC delays in our p-i-n/n-i-p PDs (see Supplementary Fig. 9). The variability among time constants suffices the demonstration purpose here, but can be reduced by optimizing the testing conditions (e.g., higher $\Delta P_{light}$, see Supplementary Fig. 10).

## Parallelizing event detection with arrays of computing units

Leveraging the capability of single CUs, we take one step further to parallelize such in-sensor event detection at the array level.

Specifically, we built a 2-by-2 cross-barred CU array composed of four CUs (Fig. 3a); these CUs (U11, U12, U21, and U22 labeled in Fig. 3b) are routed to 2 column- and 2 row-connecting lines, leaving a total of 8 G1- and 8 G2-contacts that are independently addressable (Supplementary Figs. 11 and 12).

To test the array performance, we ground the Si substrate to mitigate capacitive coupling from one CU to the other. Four CUs are gated to output the same amplitudes of $V_{out}$ under the same light condition to calibrate out the fabrication variation (see $V_p$ values in Supplementary Tab. 1). Next, we apply site-selective light illumination to the array by two independent LEDs (Fig. 3c): a 530 nm LED illuminating U11 only ($t_{on}/t_{off}$ = 200/100 ms, three 20-pulse periods, i.e., local events) and a 595 nm LED illuminating all four CUs ($t_{on}/t_{off}$ = 100/200 ms, three 20-pulse periods, i.e., global events). We then test the array operation with 3 illumination conditions (Fig. 3d–f): condition I [II] aligns the falling [rising] edges of both events; condition III applies constant global illumination.

Our experimental data under condition I [II] (taking U11 and U12 as two representative CUs. Figure 3g, h) show three specific features. First, U11 outputs 3 spikes per pulsing period corresponding to both local and global events, whereas U12 only outputs 2nd and 3rd [1st and 2nd] spikes corresponding to global events (see labels in Supplementary Fig. 13).

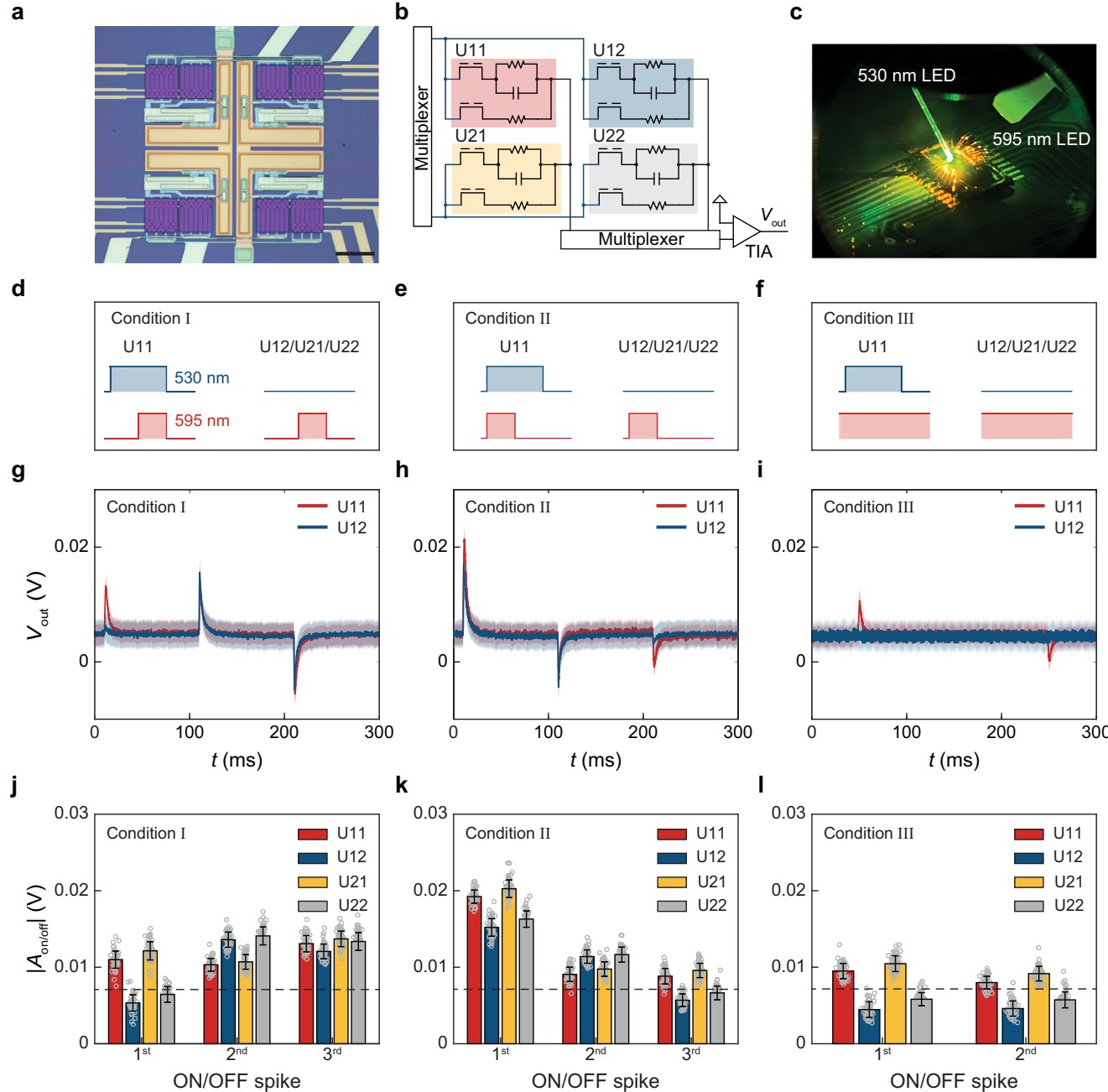

**Fig. 3 | Parallel in-sensor event detection using photodetector arrays. a** Optical image of a CU array. Scale bar, 100 μm. **b** The equivalent circuit of a CU array. **c** Optical setup with two fiber-coupled LEDs. **d** Illumination condition I. **e** Illumination condition II. **f** Illumination condition III. **g** $V_{out}$-traces of U11 and U12 under illumination conditions I. **h** $V_{out}$-traces of U11 and U12 under illumination condition II. **i** $V_{out}$-traces of U11 and U12 under illumination condition III. **j** $|A_{on}|$ and $|A_{off}|$ extracted from (**g**). **k** $|A_{on}|$ and $|A_{off}|$ extracted from (**h**). **l** $|A_{on}|$ and $|A_{off}|$ extracted from (**i**). Shaded areas in (**g–i**), and error bars in (**j–l**) both represent ±1 S.D. from a total of 60 pulses. Dash lines in (**j–l**) represent the noise level (3 S.D.) calculated from the baseline data of four CUs.

Second, the amplitude of 3rd [1st] spike from U12 is less than that from U11, in which both local and global events are switched off [on]; in contrast, their 2nd spikes have identical amplitudes since the same global events are applied to U11 and U12. Third, under condition III, U12 detects no spikes (as expected) since no $\Delta P_{light}$ is applied here (Fig. 3i). Together, these results showcase the capability of our array for parallelized in-sensor processing of site-selective events; the output amplitudes of four CUs are reliable across all pulsing periods and can be tuned by different $V_p$ values (Supplementary Fig. 14).

As the key figure-of-merit in array operation, we next examine the crosstalk among these four CUs by quantifying their spike amplitudes in the following (Fig. 3j–l):

For U12 and U22, we observe an insignificant level of crosstalk from U11 (likely from the light leakage from local events). Specifically, their 1st [3rd] spikes under condition I [II] (when local events are on [off]), as well as their two spikes under condition III, are all insignificant compared to the noise level from the baseline (Methods); their 3rd [1st] spikes under condition I [II] are smaller than those from U11 since only global events are applied.

For U21, we observe nontrivial crosstalk from U11 in all 3 conditions, where U21 and U11 have nearly identical amplitudes of all spikes. This crosstalk is found to be related to the way U11 and U21 are connected in the crossbar structure, which is reaffirmed by further experiments (Supplementary Fig. 15) and likely attributed to the signal

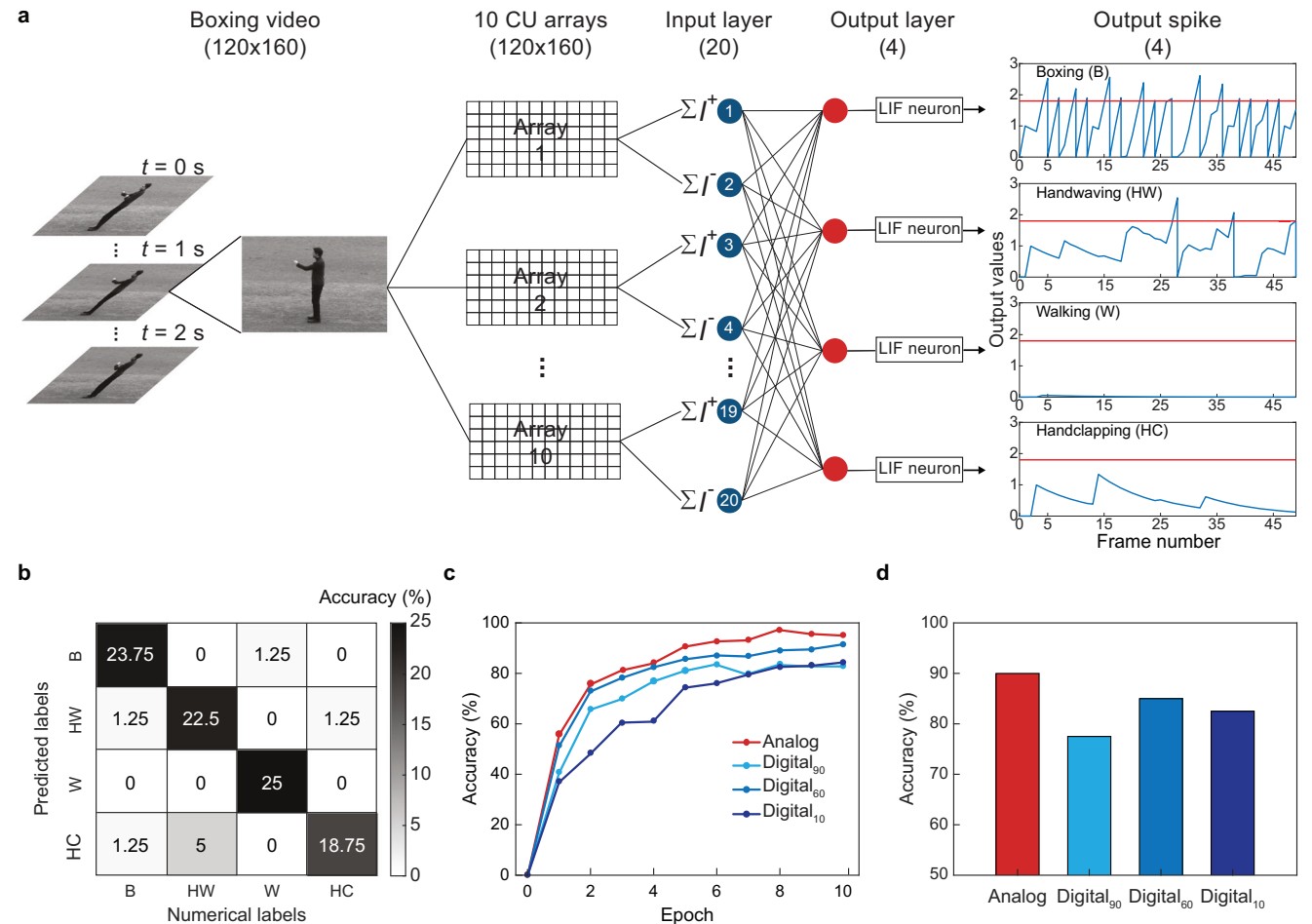

**Fig. 4 | Promise of bipolar analog readout of a CU array in classifications of human motions. a** Employing 10 CU arrays for in-sensor processing of 2-s video clips on human motions, whose bipolar analog output is fed into a SNN with a 20-node input layer ($\Sigma I^+$ and $\Sigma I^-$ from 10 arrays) and a 4-node output layer with LIF neurons to classify detected motions. **b** Confusion matrix showing predicted labels across videos in the validation set versus their numerical labels. **c** Classification accuracies over the training epoch using the analog processing method (red) or digital processing methods with a threshold of 90, 60, and 10 (light, medium, and dark blue). **d** Classification accuracies across videos in the validation set for methods in (**c**).

coupling through the parasitic gate-to-source and gate-to-drain capacitors in PDs (Supplementary Fig. 16). Nonetheless, this crosstalk can be mitigated by adjusting the gating conditions and/or array structures. In the former, we find that reversing the polarity of $V_p$ in U21 while keeping $V_p$ values in U11 will reduce the crosstalk by *ca.* 90% (Supplementary Fig. 17) at the expense of lower temporal resolution in array operation. In the latter, we can place 1-by-*m* arrays in parallel with *m* being the number of columns, where CUs from different rows are physically separated from each other to eliminate the crosstalk (Supplementary Fig. 18).

Importantly, our data suggest that the in-sensor CU array can indeed parallelize analog event detection. Such capability is achieved with zero static power because we short-circuit the branches in all CUs, and with a > 30% filling factor (FF) due to our compact modular design (Fig. 1). For these reasons, our CU array may represent a compact, low-power event detection technology.

**Motion classification with arrays of computing units**
Leveraging array-level performance and optimization steps discussed above, we now numerically examine if these CUs can form large-scale arrays to recognize human motions in sophisticated environments (Fig. 4a). First, we format a total of 100 grayscale, 2-s video clips from the KTH Action dataset (120 × 160 pixels, 50 frames) based on 25 human subjects making 4 motions (walking, boxing, hand waving, and

hand clapping with W, B, HW, and HC as the abbreviations, respectively) in a variety of scales and lightning conditions. Next, we randomly select 80 of these videos to form a training set (the rest forms a validation set), each of which is fed into 10 parallel 120-by-160 CU arrays for analog visual processing (Methods). The output of these arrays is emulated by frame differencing, leading ten 120 × 160 matrices that contain both positive and negative values (i.e., the event detected by each CU). We separately sum positive and negative values in each matrix to form a 20-node input layer of a SNN, whose 4-node output is fed into four leaky integrate-and-fire (LIF) neurons to analyze the motion in each frame and finally decide the classification result of the video. The supervised training process of our SNN offers the weight of each CU in 10 arrays, which can be achieved by setting the $R_{ph}$ values of each PD pair via gating (reverse to each other and both are proportional to the weight, Supplementary Fig. 19).

Notably, our SNN trained by bipolar analog output of CUs is able to classify motions in all 20 validation videos with 90% accuracy (Fig. 4b), outperforming those trained by digital processing methods that map frame differences into binary values through thresholding (77.5–85% in Fig. 4c, d). This result highlights the advantage of CU-based parallelized analog in-sensor computing in preserving the sub-threshold details that can be lost in digital processing approaches, thereby holding promise to recognize temporal visual information in real-world settings.

## Edge detection with single kernels

In parallel to efforts made on CU arrays for event sensing, we also examine if PD-based arrays could offer parallelized in-sensor processing of spatial visual information, showcased here for edge detection in static images. This image processing step can be achieved in electrical/optical domains[17,22,32,48,49], but are often challenged by data-intensive computing, excess power/area overhead, or low fabrication tolerance. To this end, in-sensor convolutional filtering via PD-based kernels holds promise to circumvent these limitations, whose multiply-accumulate computations (i.e., MAC) mimic the way the human retina uses to extract edge information[9,11,18,27,29]. Yet, this strategy has been mainly tested by serial scanning of a single kernel across the image, lacking parallelism needed for large-scale edge detection[33]. To address this unmet need, we first parallelize 3-by-3 PD arrays as one in-sensor kernel and examine its performance on edge detection (Fig. 5a and Supplementary Fig. 20). Specifically, we common S- [D-] contacts of 9 PDs to a testing pad; 9 G1- and 9 G2-contacts across the array are routed to 18 independent testing pads. By gating 9 PDs with different $V_p$ values (i.e., programming their $R_{ph}$), we measure the sum of their $R_{ph}$-programmed $I_{ph}$ as the kernel readout ($\Sigma I_{ph}$), which is then fed to the TIA for a voltage output $V_{out}$ (Fig. 5b).

To demonstrate in-sensor edge detection, we configure the kernel as a horizontal Prewitt filter by programming the $R_{ph}$ of 3 columns of PDs – C1 (blue), C2 (white), and C3 (red) in Fig. 5c – to be negative, zero, and positive, respectively (Supplementary Tab. 2). This configuration will result in a non-zero kernel output when there is a gradient of $P_{light}$ applied to these three columns, thereby detecting the edges of an object. Next, we move a light spot through the aperture of a microscope (dimension ~ 250 μm, $P_{light}$ = 530 mW cm$^{-2}$ at 550/15 nm) sequentially across the C1-C3. Along this trajectory, the kernel experiences a change in the gradient of $P_{light}$; its readout $V_{out}$ is collected at a step of *ca.* one-third of one column width (~ 23 μm). Consequently, we observe the following trends in the data (Fig. 5d). First, $V_{out}$ starts to decrease ($V_{out}$ = 0 when the kernel is in the dark) as the light spot enters C1, since the right edge of the light spot induces negative $I_{ph}$. Second, $V_{out}$ stays at the negative maximum until the light spot enters C3, since the light illumination on C2 generates zero $I_{ph}$. Third, $V_{out}$ increases back to zero as the light spot enters C3, since the light illumination on C3 generates positive $I_{ph}$. Fourth, $V_{out}$ stays at zero until the light spot leaves C3, since $\Sigma I_{ph}$ from all 3 columns is canceled out to be zero (absence of edges). Thereafter, due to similar reasons, $V_{out}$ increases from zero when the light spot leaves C1, stays at a positive maximum until the light spot leaves C2, and decreases back to zero when the light spot leaves C3. In sum, these results clearly demonstrate the edge detection of a horizontally-moving light spot, and we can similarly detect the edges of a vertically-moving light spot by reprogramming the kernel into a vertical Prewitt filter as well (Supplementary Fig. 21).

## Parallelized edge detection with a kernel array

After demonstrating edge detection at single-kernel levels, we extend our studies to parallelized edge detection with a kernel array, which may prove beneficial for time/data-intensive applications (e.g., autonomous driving, medical imaging). Specifically, we take the aforementioned 18-gate kernel as the functional unit to build an 8-by-8 cross-barred kernel array (composed of 576 PDs). In this array structure (Fig. 5e and f), 64 kernels are routed to 8 column- and 8 row-connecting lines; all kernels share the same 18 gate control (e.g., common a total of 64 G1-contacts from the PDs placed in the 1st row and 1st column of each kernel) by gate routing layers underneath the PDs. The resulting 100% yield array shows uniform photoresponse across 64 kernels (Supplementary Fig. 22) and is connected to off-chip multiplexers for parallel readout.

To demonstrate parallel in-sensor edge detection, we configure all 64 kernels in the array first as a horizontal Prewitt filter and then as a vertical Prewitt filter. The heat maps (i.e., contour plots) of the array readout under these two filters ($V_{H/V}$) are sequentially squared, summed, and square-rooted to generate a combined map ($V_C$) that highlights the edges detected along both directions (Fig. 5g, h). Accordingly, we test the array performance by a light spot in the shape of the aperture in a microscope (diameter ~ 300 μm), which fully illuminates 1 kernel in the center and partially illuminates 8 adjacent kernels. The resulting $V_C$ map (subtracted by values measured in the dark, Fig. 5g) shows non-zero $V_C$ values from these 8 adjacent kernels, and a zero $V_C$ from the centered kernel, correctly marking our expected edge positions of the light spot. It is again noted that both $V_H$ and $V_V$ values increase with $P_{light}$ (Supplementary Fig. 23), reaffirming the linearity of our PD-based arrays for analog in-sensor edge detection. Taking one step further, our array is also able to simultaneously detect the edges of multiple objects, provided here by two cell-like light spots defined by a shadow mask (Fig. 5h).

In sum, these results suggest that our kernel array is able to achieve parallel detection of the edges in both single and multiple objects. Notably, our array consumes zero electrical power due to the short-circuited operation, and features a > 90% FF due to the compact modular design. Therefore, these arrays could be viewed as a viable low-power edge-detection technology that can be built into an integrated chip form.

## Classification of handwritten digits with kernel arrays

Encouraged by the success of parallelized edge detection at array levels, we take a step further to numerically examine if array-based convolutional filtering can extract details needed to classify objects in the image (Fig. 6). First, we reshape 70,000 grayscale images of handwritten digits (0-9) from the MNIST database into 24 × 24 matrices (Methods), and randomly select 60,000 [10,0000] of them to form a training [validation] set. Next, we sequentially configure all 64 kernels in the 8-by-8 kernel array into 10 convolutional filters (i.e., CF) via gating, whose $R_{ph}$ values can be decided by the weights of filters learned from the training process (see discussions below and data listed in Supplementary Fig. 19). The array at each configuration filters a training image into an 8-by-8 processed image with bipolar analog values. The resulting 10 processed images are then flattened into the 640-node input layer of an ANN, whose 10-node output is used to decide the classification results of the training image (Fig. 6a). Our kernel array in this case offers the high-dimensional input of the ANN thanks to its reconfigurability via gating.

Notably, our ANN trained by bipolar analog output of the kernel array can classify digits across 10,000 validation images with 94.8% accuracy (Fig. 6b), on par with a convolutional NN (CNN) based on scanning single kernels across the images (97.4%, 4840-node input) and outperforming a NN trained by original images (89.6%, 576-node input, Fig. 6c, d). This result suggests the advantage of kernel-based parallelized analog in-sensor computing in cutting the compute overhead (vs. CNN) and increasing the weights of key spatial details in the image (vs. original images), thereby holding promise to recognize spatial visual information with low hardware and power budget.

## Discussion

We have presented two scalable in-sensor visual processor arrays based on a compact modular design (Supplementary Tabs. 3 and 4) of α-Si based dual-gate PDs for parallelized analog processing of temporal and spatial visual information (e.g., events and edges), respectively. The uniformity of PDs in our array is on par with prior wafer-level statistics[29] (Supplementary Fig. 22), and can further be controlled via gate tuning (Supplementary Tables 1 and 2). Both arrays consume zero static power at device levels and share the CMOS compatibility that lends themselves for large-scale visual processing tasks with high parallelism. By programming the photoresponse of independent PDs in these arrays, we are capable of parallelized analog processing of site-

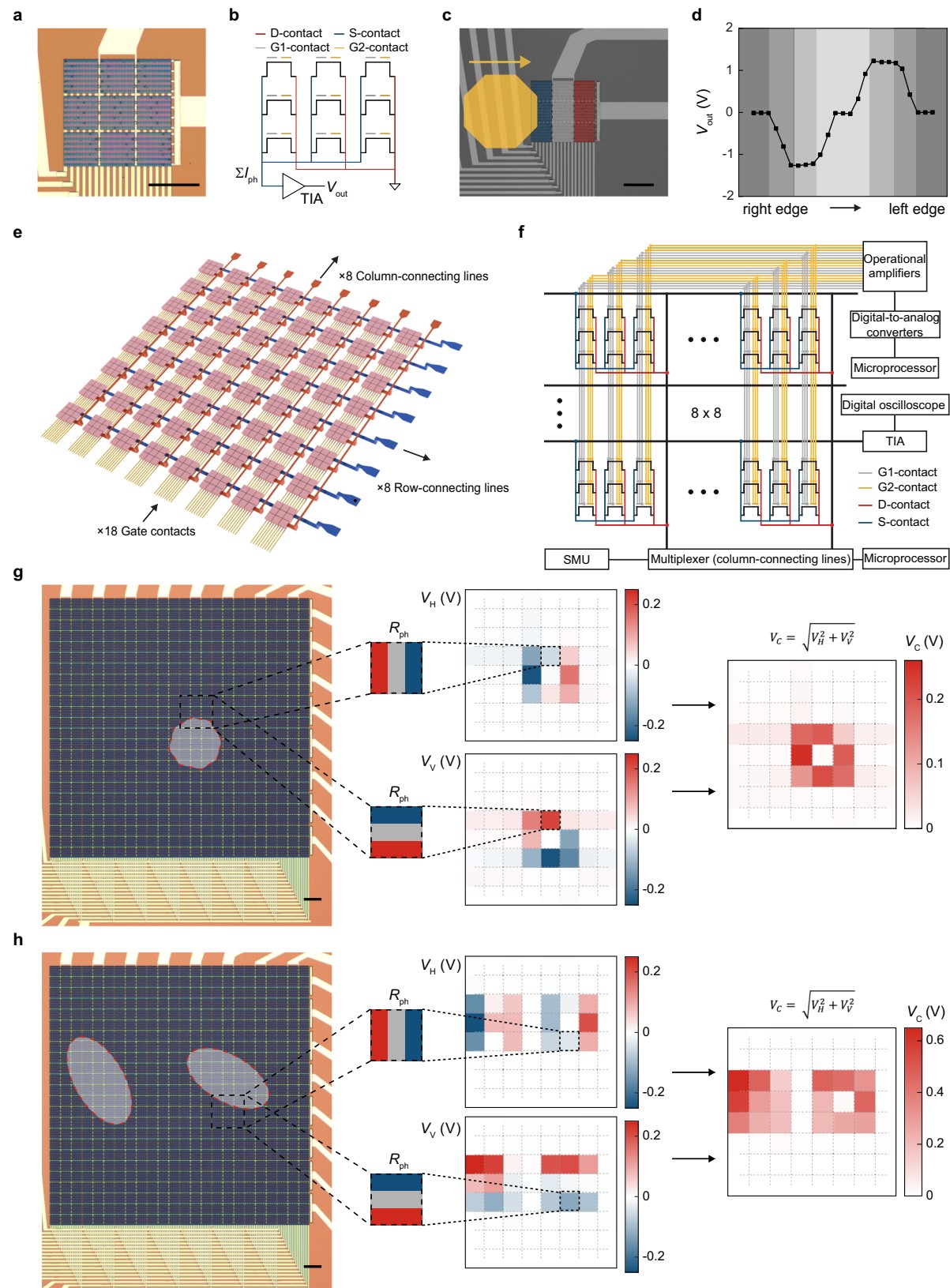

specific events at sub-ms precision and edge profiles of multiple objects, leveraging their scalability, uniformity, and strategies to mitigate the crosstalk. Furthermore, we numerically demonstrate that our CUs can be scaled up to multiple large-scale arrays that can work together to classify human motions with 90% accuracy, whereas our kernel array can be reconfigured to multiple CFs that can provide high-

dimensional input to a SNN to classify handwritten digits with 94.8% accuracy. Such an array-level of analog in-sensor visual processing may shed light on smart sensing systems aimed at large-scale, data-intensive, and latency-sensitive computer vision tasks.

Moving forward, our analog in-sensor visual processor arrays can add to the advancement of multifunctional computer vision hardware

**Fig. 5 | Gate-tunable analog in-sensor edge detection at single-kernel and kernel-array levels. a** Optical image of a single kernel. **b** The equivalent circuit of a single kernel. **c** A kernel configured as a horizontal Prewitt filter is used to detect the edges of a horizontally moving light spot ($P_{light}$ = 530 mW cm$^{-2}$ at 550/15 nm). **d** $V_{out}$ values measured with the light spot in (**c**) horizontally moving at a *ca.* 23 μm step. **e** The schematics of an 8-by-8 kernel array. **f** The circuit employed to configure the kernel array for parallelized edge detection (SMU: source-measurement unit).

**g, h** Parallel in-sensor edge detection of one aperture-defined light spot (**g**) and two shadow-mask-defined light spots (**h**) using the kernel array ($P_{light}$ = 530 mW cm$^{-2}$ at 550/15 nm). The combined map of $V_C$ across the array are calculated from the maps of $V_H$ and $V_V$ measured when all 64 kernels are configured as a horizontal and a vertical Prewitt filter, respectively (**g, h**). To measure the map of $V_H$ [$V_V$], the red, gray, and blue column [row] of each kernel is gated to have positive, near-zero, and negative $R_{ph}$ values, respectively. Scale bars in (**a, c, g,** and **h**), 100 μm.

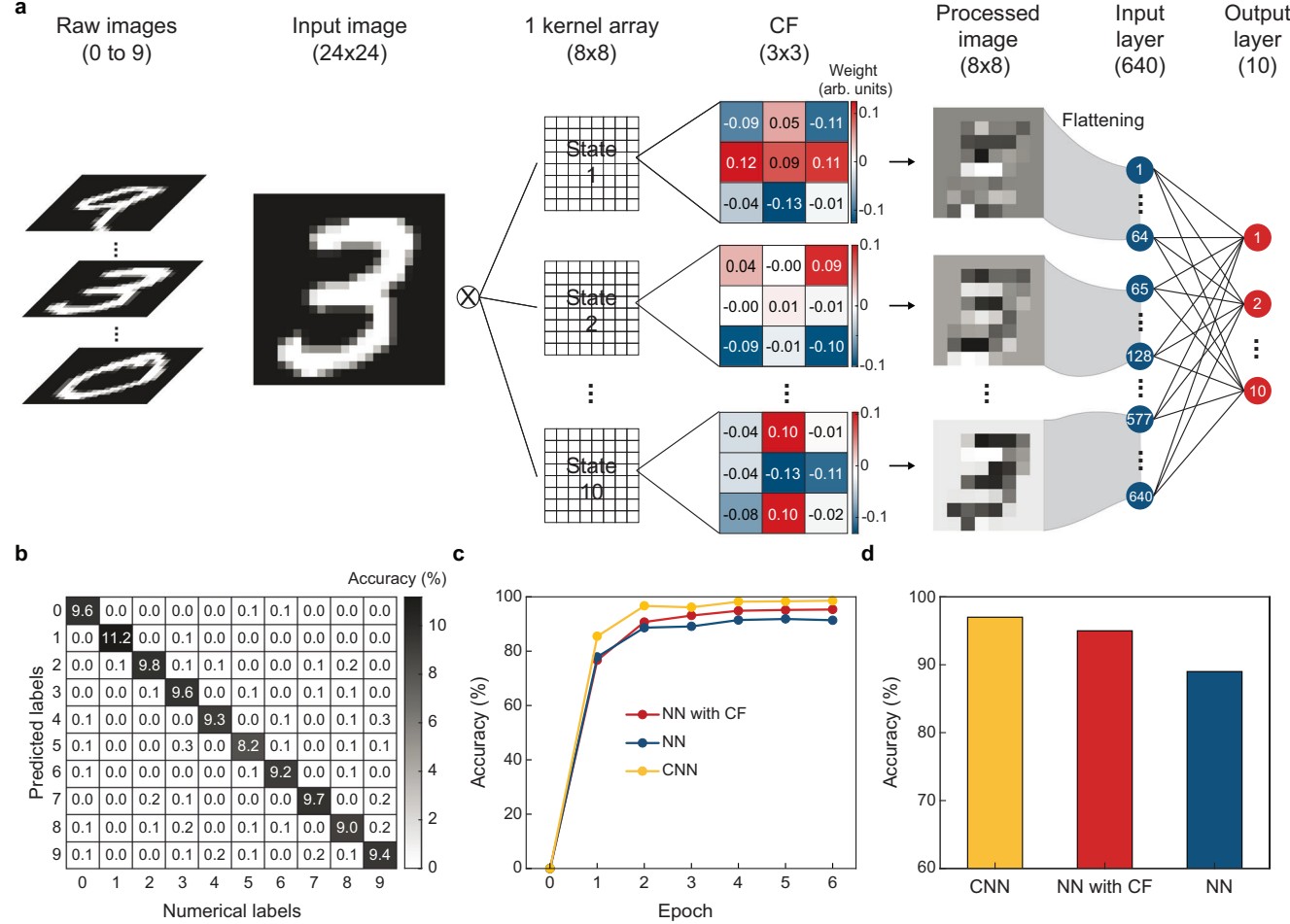

**Fig. 6 | Promise of bipolar analog readout of a kernel array in classifications of handwritten digits. a** Sequentially configuring 64 kernels in a kernel array into 10 CFs (i.e., forming 10 states of the same array) that can filter one input image into ten 8-by-8 processed images, whose bipolar analog values are fed into an ANN with a 640-node input layer and a 10-node output layer to classify detected digits. **b** Confusion matrix showing predicted labels across images in the validation set

versus their numerical labels. **c** Classification accuracies over the training epoch using NNs trained by parallelized convolutional filtering with a kernel array (red), convolutional filtering with a single kernel scanning across the image (yellow), and non-filtered image (blue). **d** Classification accuracies across images in the validation set for methods in (**c**).

that can process visual information with ultralow power consumption and high spatiotemporal resolutions. Future studies combining zero-biased PDs/CUs and low-power readout circuits may prove beneficial to build low-power edge systems (e.g., mobile platforms) for intelligent computing. Moreover, the CMOS compatibility of our arrays may allow them to be monolithically integrated with analog in-memory computing devices[6,23,24,27,30,46,47] (other CMOS-compatible or low-dimensional materials may also be chosen to build application-specific arrays in the future), which can form a fully-integrated on-chip analog deep-learning neural network to offer near-real-time sensing, processing, and recognition of the visual targets[50,51]. This integration approach could pave new ways in a broad range of machine vision applications, especially in scenarios that demand simultaneous processing of spatiotemporal information (e.g., biomedical imaging[3,4] and autonomous driving[1]). For instance, our fully integrated system

may enable efficient extraction of the spatial attributes of cells, tissues, and organs (e.g., size, shape, location), and fast tracking of their dynamic activities with biological[52] or medical[53] significance (e.g., Ca$^{2+}$ fluxes, blood oxygenation). On the other hand, our technology may empower human-computer interaction applications (e.g., augmented reality, virtual reality)[54] and automated navigation systems[1,50,55] that heavily rely on timely extracting both spatial information (e.g., target recognition) and temporal dynamics (e.g., the motion of fast-moving objects).

Finally, we remark a few steps to further optimize the performance of our in-sensor processor arrays. First, to mitigate the electrical crosstalk across the array, our event-detecting CUs could be built into multiple rows of 1D arrays (instead of a cross-barred array). Second, both CUs and convolutional kernels could be connected to selectors (e.g., switching transistors) in series to mitigate the sneaky current

paths and avoid in-sensor computation errors; such selectors could be integrated underneath PDs with no area penalty. Third, the temporal resolution of our CU array ($t_{rise/fall}$) can be further improved by optimizing RC values and the upper limit of our TIA bandwidth. A sub-μs resolution of event detection can be achieved by CU arrays integrated with smaller Rs and Cs, and those wired to high-bandwidth TIA circuits; such smaller RC values can also benefit the reduction of heat dissipation of the circuit. Fourth, the compactness of our CU array can be further improved by stacking PDs on top of the RC elements, choosing smaller Rs and Cs[40], integrating on-chip TIAs for each row of CUs, or using multilayer metallization to reduce the areas occupied by gate lines.

## Methods

### Device fabrication

In this work, sputtered Ti/Pt layers (10/50 [100] nm) are chosen to form G1-, G2-, S- and D-contacts, gate-routing lines, top electrodes of the C, and connection lines [wire-bonding pads]; evaporated Cr/Au layers with a thickness of 10/300 nm [10/50 nm] are chosen for vias [the bottom electrode of the C]; a 300 nm PECVD-SiO$_2$ layer is chosen to act as the passivate layer; a 30 nm ALD-Al$_2$O$_3$ [15 nm ALD-HfO$_2$] layer is chosen as the gate oxide layer [dielectric layer for the capacitor], respectively; and a 250 nm [100 nm] PECVD-based intrinsic [n-doped] α-Si layer is chosen to act as the light-absorbing region [integrated Rs].

For single PDs (Supplementary Fig. 1), we first pattern gate-routing lines on top of a SiO$_2$/Si substrate[56] (with a *ca.* 300 nm SiO$_2$ layer thermally grown on top of a p-doped Si substrate) and cover them with a passivation layer. Next, we form vias through the passivation layer by reactive ion etching (RIE) and metallization steps; G1- and G2-contacts and their testing pads are then deposited on top of the vias to make connections. On top of them, we next sequentially form a gate-oxide layer[29], S/D-contacts together with their testing pads, and intrinsic α-Si regions for light absorption (patterned via RIE). Finally, we passivate the device and use RIE steps to open four testing pads that connect to G1-, G2, S-, and D-contacts.

For single CUs (Supplementary Fig. 6), we first form an integrated C by sandwiching a HfO2-based dielectric layer between a top electrode and a bottom electrode on top of a SiO$_2$/Si substrate. The top electrode is formed together with four gate-routing lines, from which we later build two identical PDs using the aforementioned steps. Different from single-PDs, though, here we form vias not only on gate routing lines (serve to later connect to gate contacts and their testing pads), but also on the top electrode of C (serve to later connect to two integrated Rs). Also, the S- and D-contacts of PDs are formed together with connection lines, which serve to wire the C, Rs, and PDs later as a 2PD-2R-1C circuit. Last but not least, after patterning the intrinsic α-Si regions of PDs, we pattern two Rs from an n-doped α-Si film (by RIE) on top of their pre-formed connection lines to complete the CU. Afterwards, we passivate the CU and use RIE steps to open the bottom electrode of C, the S-contacts of two PDs, and the four testing pads that are connected to their gate contacts. We then conduct a final metallization step to form connecting wires and testing pad (serve to connect to the bottom electrode of C and S-contacts), and metal features right above the two Rs (i.e., light blockers) to avoid light-induced resistance change.

For the CU array (Supplementary Fig. 11), we form four identical CUs with the aforementioned steps. Different from single-CUs, though, here we make a common connection between the bottom electrodes of Cs in two CUs on the same column (U11 + U21, U12 + U22). We then passivate the device and use RIE steps to open S-contacts of CUs and bottom electrodes of Cs. Thereafter, we conduct a final metallization step to form light blockers, and wire S-contacts [bottom electrodes of Cs] to the row-[column-] connecting lines.

For single kernels (Supplementary Fig. 20), we form 9 PDs placed in a 3-by-3 array (a total of 18 independent gate contacts) using the

same steps as single PDs. Next, we passivate the device and use RIE steps to open the testing pads that are connected to 18 gate contacts, as well as the S- and D-contacts of all 9 PDs. We then conduct a final metallization step to common 9 S- and 9 D-contacts via connecting wires and two testing pads, respectively.

For the kernel array (Fig. 5e, f), we form 64 identical kernels placed in an 8-by-8 array with the aforementioned steps. Different from single-kernels, though, here we first common 18 independent gate-routing lines from 8 kernels in each column, and further wire the resulting 144 gate-routing lines to 18 global gate contacts (using vias, connecting wires, and the corresponding testing pads) that simultaneously control $V_p$ values of all 64 kernels. Moreover, we common the S-contacts of 8 kernels in the same row, leading to a total of 8 independent row-connecting lines; in contrast, the D-contacts of all 64 kernels are still separated. Afterwards, we passivate the device and use RIE steps to open the testing pads that connect to 18 global gate contacts, the 8 row-connecting lines, and the 64 D-contacts. We then conduct a final metallization step to common 8 D-contacts in each column and wire them via 8 column-connecting lines and their testing pads, as well as connecting 8 row-connecting lines (wired to S-contacts) to 8 testing pads, respectively.

### Device characterization

Single PDs are fully characterized by a semiconductor device parameter analyzer (Keysight B1500A), which serves to offer the biases of their G1-, G2-, S- and D-contacts via four independent manipulators.

Single CUs or the CU arrays are first wire-bonded onto a loading printed circuit board (PCB, see Supplementary Fig. 24), which is then wired to a gating PCB and a multiplexing PCB (both PCBs are powered by a power supply, Keysight E3631A). The gating PCB offers 18 independent gate biases via microprocessor-controlled (ardATmega328) digital-to-analog convertors (MCP4822) and 18 operational amplifiers (LF356, offset and amplify the output range); the microprocessor is set to gradually ramp $V_p$ values at 0.15 V s$^{-1}$ to avoid a large transient current that may cause oxide breakdown. The multiplexing PCB, on the other hand, serves to select the CU (either single CUs or one CU from the CU array) by two multiplexers (TI ADG419) that are controlled by another external microprocessor. For each selected CU, we bias the S-contacts of 2 PDs at 0 V with a source-measurement unit (SMU, Keysight 2902 A), and connect the bottom electrode of C to the positive input of a TIA (Stanford Research System SR570, high bandwidth mode, gain = 2 × 10$^8$ V A$^{-1}$), whose negative input is biased at 0 V to convert the short-circuited branch current to $V_{out}$ values. The TIA output is then fed into a Hum bug noise eliminator (A-M Systems) to remove the 50/60 Hz noise, followed by a digital oscilloscope (Pico4824) to sample the filtered $V_{out}$ traces at 10 kHz.

Single kernels or the kernel arrays are first wire-bonded onto another loading PCB, which is then wired to the gating PCB (the same one used for CUs) and another multiplexing PCB that are powered by the same power supply. The gating PCB is operated the same way as the CU testing to offer 18 independent gate biases. On the other hand, we use one multiplexer (TI ADG405) on the multiplexing PCB to select the column-connecting line of the select kernel (note: single kernels are viewed as a 1-by-1 array here) via the external microprocessor and bias it at 0 V via the SMU; the row-connecting line of the select kernel is connected to the TIA (low noise mode, gain = 2 × 10$^9$ V A$^{-1}$), followed by the digital oscilloscope to sample the $V_{out}$ traces at 1 kHz.

### Optical setup

During our experiments, we use an upright microscope (Nikon FN1) equipped with a Zyla4.2 plus sCMOS (scientific complementary metal-oxide semiconductor) camera (Andor, USB 3.0) and a SPECTRA X light engine (Lumencor) to take device images, align the light spot to the device, and provide 550/15 nm illumination patterns through a CFI60 Plan Achromat 10 × objective lens (NA = 0.25, Nikon). We also use two

fiber-coupled LED (Thorlabs, M539F2 and M595F2) to provide 530 nm and 595 nm illumination, respectively.

We test individual PDs in a CU by spatially confining the 550/15 nm illumination patterns (see Supplementary Figs. 7 and 15). For the testing of CU arrays, we apply the 530 nm LED illumination to U11 only as local events, and the 595 nm LED illumination to all CUs as global events (Fig. 3). For single-kernel experiments (Fig. 5c, d) and the kernel-array experiments in Fig. 5g, we shape the light spot by the aperture of the microscope. For the kernel-array experiments in Fig. 5h, we place a shadow mask – made of Pt/SU8 layers (300 nm/2 μm) patterned on a coverslip – face down onto the kernel array; this way, we are able to illuminate two separate regions on the array. Among these experiments, we decide the value of $V_p$ needed to output targeted $V_{out}$ values at the steady state of the select CU/PD (Supplementary Fig. 25).

## Circuit simulation
We conduct circuit simulation under the LTspice environment. Specifically, we model the simplified equivalent circuit of PDs with a current source, an ideal diode, a junction resistor ($R_p$), a junction capacitor ($C_j$), a series resistor ($R_s$), a parasitic capacitor existing between G1- and S-contacts ($C_{G1-S}$), and a parasitic capacitor existing between G2- and D-contacts ($C_{G2-D}$)[57]. In this work, we set these resistance values in two different approaches. In the first approach, $R_p$ is estimated from the slope of the $I_S$ - $V_S$ curve at $V_S = 0$ (measured in dark, Fig. 1b)[58], while $R_s$ is neglected by assuming $R_s \ll R_p$. In the second approach, both $R_p$ and $R_s$ are estimated from the $I_S$ - $V_S$ curve in Fig. 1b using the Shockley model reported before[35]. On the other hand, the value of $C_j$ is estimated from a quasi-static capacitance-voltage curve (QSCV, by B1500A) measured between S- and D-contacts of the PD with $V_p$ being biased at 2.5 V (by gating PCB). Finally, the value of $C_{G1-S}$ [$C_{G2-D}$] is estimated from the QSCV curve measured between G1- and S-contacts [G2- and D-contacts] of the PD, with the G2- and D-contacts [G1- and S-contacts] being floated.

## Data analysis
The values of $A_{on/off}$ are obtained by subtracting the positive/negative maximum of the $V_{out}$ traces by the 10-point average of the baseline data from the 1 ms window right before each light pulse. The noise level in Fig. 3j–l [Supplementary Fig. 14c, d] is defined as 3 times the S.D. in the baseline (measured from 1-s data right before the first light pulse and averaged by four CUs [averaged when each CU is biased at two different $V_p$ values] in the array).

## Simulations on the CU array for classifying dynamic motions
We first select a subset of the 13-s video clips in the KTH Action dataset (25 frames per second, 120 × 160 pixels) to include four types of human motions (walking, boxing, hand waving, hand clapping). A total of 100 videos (4 motions by 25 human subjects) were recorded in a variety of scenarios: outdoors, outdoors with scale variation throughout the video, outdoors with different clothes, and indoors, which are ideal to examine the performance of motion detection in a complex environment. To examine if our arrays can classify these motions in a timely manner, we next tailor these videos into 2-s video clips (50 frames) for our analysis, detailed as follows. For videos in the category of walking, we delete low-content frames when the human subject is out of the scope, and select the first 50 remaining high-content frames. For other non-walking videos, we select the first 50 frames of data (as the human subject is always in the scope). Afterwards, we randomly select 20 out of 25 formatted videos (50 frames each) in each category of motions (80 in total) as the training set to build the spiking neural network (SNN) model[2,28]; the remaining 20 formatted videos form the validation set.

On the device end, we employ ten 120 × 160 CU arrays (in the form of 120 parallel 1-by-160 arrays as discussed in Fig. 3) to process each 2-s video in parallel to maximize the classification performance (see below). The photoresponsivity of the PD pair in each CU (a total of 120 × 160 × 10 CUs in ten CU arrays) is determined by the SNN modeled by MATLAB Deep Learning Toolbox (see Supplementary Fig. 19) and can be experimentally achieved by gating. The output of each CU array is emulated by subtracting the grayscale values in each frame by those in the previous frame (i.e., frame difference), yielding a 9-bits 120 × 160 matrix that contains both positive and negative values (i.e., the photocurrent detected at each CU) to represent the analog visual processing capabilities of CUs. Afterwards, the 20 summed positive and negative values in the 10 matrices ($\Sigma I^+$ and $\Sigma I^-$ that can be achieved by off-chip electronics) will form a 20-node input layer of the SNN. The output layer of the SNN is composed of 4 nodes, whose numerical values represent the probability of each frame being classified into 4 classes of motions (via the SoftMax function in MATLAB). These output values are followed by LIF neurons to output the real-time spiking waveforms (a leaky constant is set to be the default value 10, a firing threshold is set to be 1.8 when the output value equals to 1 for two consecutive frames). The output node that shows the greatest number of above-threshold spikes will be taken as the predicted label of the input video. Finally, the prediction accuracy is defined as the fraction of correct predictions of 20 videos in the validation set using our trained SNN model, whereas the confusion matrix is defined as the percentage of each predicted label versus each numerical label out of 20 validation videos.

For comparison purposes, we also conduct the same motion classification task using digital visual processing strategies, in which case the frame difference is digitized into +1, 0, and −1 with different thresholds before conducting the $\Sigma I^+$ and $\Sigma I^-$ operation. Specifically, our 2-s videos have grayscale values ranging from 0 to 180, thereby, the pixels in the 120 × 160 matrix of the frame difference ranges from −180 to 180. For this reason, we choose half of 180 (90), one third of 180 (60), and a small number (10) as three thresholds for comparative studies. The pixel values larger [less] than the positive [negative] threshold will be digitized as 1 [−1], those between positive and negative thresholds will be digitized as 0.

## Simulations on the kernel array for classifying static images
We first trim grayscale images of handwritten digits 0-9 in the MNIST database from 28 × 28 into 24 × 24 matrices by removing two rows or two columns of data in the upper, lower, left, and right sides of the original image (i.e., near-zero values). A total of 60,000 trimmed images are randomly selected as the training set to build the neural network (NN) model with the array being reconfigured into different states (i.e., acting as convolutional filters); the remaining 10,000 trimmed images form the validation set. On the device side, our kernel array in Fig. 5 has the dimension 8 × 8 with each kernel being composed of 3 × 3 PDs. We thus sequentially reconfigure one such array into 10 different states, each of which convolutionally filters the 24 × 24 trimmed image into an 8 × 8 processed image; the 3-by-3 weight matrices of these states (determined by the NN modeled by MATLAB Deep Learning Toolbox, see Supplementary Fig. 19) represent the photoresponsivity we need to assign to the 9 PDs in each kernel. Afterwards, we use a ReLU function to zero the negative values in a total of 10 processed images, and flatten them into a 640-node input layer of the NN. The output layer of the NN is composed of 10 nodes, whose numerical values represent the probability of the trimmed image being classified into 10 classes from 0 to 9 (via the SoftMax function in MATLAB); their maximum is then used to decide the predicted label. Finally, the prediction accuracy is defined as the fraction of correct predictions of 10,000 trimmed images in the validation set using our trained NN model, whereas the confusion matrix is defined as the percentage of each predicted label versus each numerical label out of 10,000 validation images.

For comparison purposes, we also conduct the same image classification task using a pure NN model with the 24 × 24 trimmed images

directly flattened into 576 nodes in the input layer (i.e., no convolutional filtering steps), and a convolutional NN (CNN) model also built with 10 different 3 × 3 convolutional filters, which however are used to scan across the 24 × 24 trimmed images pixel by pixel to yield 22 × 22 processed images (i.e., a CNN with 4840 [10] nodes in the input [output] layer).

## Data availability

All data supporting the findings of this study are available within the article and its Supplementary Information. Any additional requests for information can be directed to and will be fulfilled by the corresponding author.

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

## Acknowledgements
The authors are grateful for the support of this research by the National Science Foundation under contracts ECCS 2046031 (G.X.), ECCS 2055457 (G.X.), and CCF 2133475 (Q.X.). We thank X. Zhang and Y. Chen for scientific discussions.

## Author contributions
G.X. and Q.X. conceptualized the research. Z.X., W. L., Q.X., and G.X. conceived and designed the experiments. M.Z., Z.X., and D.M. fabricated the devices. Z.X., W.L., and M.Z. conducted the experiments. Z.X., W.L., M.Z., and G.X. analyzed the data and wrote the paper. All authors discussed the results and reviewed the manuscript.

## Competing interests
The authors declare no competing interests.
