## [Transparent Peer Review file · Nature Communications]

Parallelizing analog in-sensor visual processing with arrays of gate-tunable silicon photodetectors

Corresponding Author: Professor Guangyu Xu

Version 0:

Reviewer comments:

Reviewer #1

(Remarks to the Author)

The authors presented a silicon-based photodetector array and discussed applications in neuromorphic camera. The chips were quite impressive and the paper was well written. However, I feel that the major highlights or novelties of this work right now may not catch the criterium of Nat. Commu.

(1) The device-level novelty of this work was somehow engineering job, rather than scientific breakthrough.

By comparing Fig.2 of this work with Fig.4 by Y. Zhu et al. (Nat. Electr., 2023, <https://doi.org/10.1038/s41928-023-01055-2>), clearly, the design of differential pair based DVS was first proposed by Zhu et al but not the authors. The authors of this work claimed that they made several significant improvements to the original design. One was to realize the resistor and capacitor on silicon chip rather than with off-chip devices on PCB.

I cannot agree with their arguments. First, silicon-based in-sensor computing device was demonstrated by H. Jang et al. (Nat. Electr., 2022, <https://doi.org/10.1038/s41928-022-00819-6>) The basic principle of the device presented in this work was the same as the above work, and of course with some improvement on the channel materials and electrode design (Fig.1 of this work). Second, comparing to Zhu's original design, to integrate the capacitor and resistor on silicon chip was hardly some scientific breakthrough. Engineeringly, it was trivial as long as you have sufficient funding.

(2) There is a severe gap between results presented in Fig.2,3 and Fig.4. Or the logic from Fig.2,3 to Fig.4 is somehow incoherent.

Fig.2 and Fig.3 showed silicon-based DVS cell and small array and the associated measurements. Therefore, from Fig.2 and Fig.3 of this work, as a reader I expected very much that the authors was going to show large-scale array of in-sensor event-detecting. Yet, in Fig.4 the array becomes the ordinary optoelectronic sensors, i.e., light intensity-sensitive, rather than light change-sensitive. And here with such array the authors showed the design of convolutional kernels for edge detection. In my opinion, edge detection was some "change" in spatial continuity, while event-driven refers to change in temporal continuity.

From Fig.3e, I understand that some crosstalk effect may hamper the large-scale realization of event-sensing cells. That may explain why in Fig.4 the authors abandoned discussion on the event-sensing and turned back to ordinary light-intensity sensing.

I see that the authors try to argue the parallel computing potential of the array shown in Fig.4. But it is somehow weird to turn from DVS in Fig.2,3 back to ordinary optoelectronic sensor in Fig4. If DVS shown in Fig.2,3 does not apply to large-scale integration, why do authors spend so much discussing it since there was nothing quite new about it?

Thus, is it possible that the authors address the crosstalk effect seriously and therefore realize the large-scale integration circuit of event-sensing cells? I feel that such engineering issue may be the real challenge on the road of neuromorphic camera and addressing it may be the real highlight.

(3) what about the algorithm-level innovation adapted to the presented analog DVS?

In Zhu's original design of differential pair-based DVS, there was no positive/negative threshold component (stepwise

functions). What they obtained was hence some kind of analog DVS rather than the digital (or to say polarity) DVS commercially available now. Personally, I feel it is hard to say such analog DVS is advantageous or disadvantageous over the digital DVS. It may depend on the algorithm-level adaptation to the analog output values of the former.

Then, from Fig.2a it seems that the authors of this work totally adopted the design by Zhu* and moreover, they outlined “analog” in the paper title. Since analog DVS is fundamentally different from the commercial DVS, could the authors provide some convincing algorithm design to justify that the “analog” circuit proposed in Fig.3 would be of some significant advantages over the digital counterpart by the commercial DVS?

*Maybe I made a mistake. Zhu et al published the Nat. Electr. work very recently, while the authors' chip should be fabricated before this timing.

Reviewer #2

(Remarks to the Author)

In this manuscript, the authors introduce two scalable in-sensor visual processing arrays based on dual-gate amorphous-silicon photodiodes for multiplexed event sensing and edge detection of visual objects. These arrays can process both temporal and spatial visual information in an analog manner, mimicking the functionality of the human retina. The concept of in-sensor computing presents a significant advancement for next-generation computer vision hardware, and this paper makes an innovative contribution to this field. The paper provides detailed technical information in the Methods and Supplementary Information sections, enabling readers to reproduce the research. Overall, I believe this manuscript is well-prepared and organized. I would like to recommend this manuscript for publication in Nature Communications. Here are some specific comments and suggestions:

- Q1. I am curious about why two different wavelengths (530 nm, 595 nm) of light were used in Fig. 3. Based on my understanding of this article, two superimposed light pulses with the same wavelength can also produce the same result.
- Q2. Please review 'at 550/15 nm' on page 13, as it confused me. Is it a typo?
- Q3. I suggest that the authors redraw Fig. 4d and 4g to make them more easily understandable.
- Q4. Can the uniformity of the PD devices or the variations in the fabrication process be well-controlled? Since the multiple PD cells work collaboratively, it is important for their performance to be similar to each other.
- Q5. The zero-bias condition ($V_S = V_D = 0$ V) is very promising for low-power edge applications. I am curious about the possibility of achieving a zero-power intelligent in-sensor computing system. Could the authors provide a discussion or outlook on this point?

Reviewer #3

(Remarks to the Author)

In this manuscript, the authors report on in-sensor visual processing arrays of dual-gate amorphous-silicon photodiodes, used for multiplexed event sensing at sub-ms precision and edge detection of multiple objects, respectively. The device arrays are capable of processing both temporal and spatial visual information. It is an important topic in the area of vision systems, requiring improved vision systems with reduced latency and improved performance. Some valuable engineering work can be seen in integrating the PD and achieving different spiking polarities in an appropriate circuit form, but it incurs area penalty that novel devices can achieve in a single device. As such, technological and scientific advancement might seem to be limited in the current version. My comments to the authors are as follows.

1. It is mentioned that a-Si is chosen for its high absorption coefficient and its ability for 3D stacking, which is important for high-density integration. However, the use of a-Si brings about problems, including defects, non-uniformity, performance, etc. Can the authors comment?
2. Can authors explain the reason for using 595 nm light in their experiment? What would the different wavelengths (visible light) have on the behavior of the PD I-V?
3. In Figure 1b, the energy band diagram of the PD is drawn. However, the gate biases applied change the regions to both n and p type. If the device is not under thermal equilibrium conditions, why would the fermi level line up?
4. In supplementary figure 4, the same bias is applied to both gates and I-V obtained and illuminated. Have the authors investigated the situation where no light is illuminated?
5. In figure 2a, the spikes are generated with the aid of passive elements and TIA, this makes a cell somewhat bulky. Such behavior has also been in some reported works on 2D materials where the events can be detected at different light intensities.
6. In figure 2, the error bar associated to the performance of the PD appears to be quite large, especially in Figure 2c (for example) between the on spike and off spikes. Can the authors comment on the reliability of using such device in large arrays.
7. It is noted that the gates in the arrays are independently addressed; would this be a limiting factor for large arrays?
8. With the current array design, can the crosstalk be eliminated with appropriate gate biases? This has not been discussed in the manuscript.
9. Especially for matured materials made devices, I think that a proper benchmark with reported state of art is important, highlighting the advantages of current work. A lot has been claimed in the discussion section on scalable in-vision processor array and functions. These have to be properly compared as well or the claim might appear to be weak.
10. The authors should demonstrate more applications regarding static image processing and event-based motion processing based on this array.

Version 1:

Reviewer comments:

Reviewer #1

(Remarks to the Author)

The authors have addressed all of my concerns and the revisions made by the authors greatly amplify the innovative aspects of their research. I suggest the publication of this work.

Reviewer #2

(Remarks to the Author)

I read the response carefully, and found that all my comments have been well addressed in the revised manuscript. I have no further questions and am happy to recommend it for publication.

Reviewer #3

(Remarks to the Author)

Comments are generally answered satisfactorily, but I have the following follow up questions

1. Authors replied on the use of laser annealing to convert amorphous Si to poly Si. I think authors need to think carefully on the reply to this comment as there is a clear difference in behavior (eg. Absorption) for both materials. What kind of material is more suited for the application presented in the work, and not just to mitigate the non-uniformity issues. Hence, this part needs to be better justified in the manuscript.
2. I hope authors are aware that 2D materials are scalable as well and added discussion does not seem to capture the differences between Si and 2D materials clearly. Otherwise it will be misleading to readers and hinder the further advancement of this technology.
3. Also, while it is true that multi-metallization can be used to address individual pixels but you cannot avoid the low Si area efficiency for arrays that needs to be addressed in this manner as the number of I/O required cannot be reduced. I think this disadvantage needs to be mentioned as well.

Point-by-Point Response to Reviewers' and Editor's Comments

A. RESPONSE TO REVIEWER #1

[Reviewer #1, General Comment]

“The authors presented a silicon-based photodetector array and discussed applications in neuromorphic camera. The chips were quite impressive and the paper was well written. However, I feel that the major highlights or novelties of this work right now may not catch the criterium of Nat. Commu.”

[Response]

We thank Reviewer #1 for this encouraging remark, and would like to delineate the novelties and impact of this work through this revision.

[Reviewer #1, Specific Comment #1]

“(1) The device-level novelty of this work was somehow engineering job, rather than scientific breakthrough.

By comparing Fig.2 of this work with Fig.4 by Y. Zhu et al. (Nat. Electr., 2023, <https://doi.org/10.1038/s41928-023-01055-2>), clearly, the design of differential pair based DVS was first proposed by Zhu et al but not the authors. The authors of this work claimed that they made several significant improvements to the original design. One was to realize the resistor and capacitor on silicon chip rather than with off-chip devices on PCB.

I cannot agree with their arguments. First, silicon-based in-sensor computing device was demonstrated by H. Jang et al.(Nat. Electr., 2022, <https://doi.org/10.1038/s41928-022-00819-6>) The basic principle of the device presented in this work was the same as the above work, and of course with some improvement on the channel materials and electrode design (Fig.1 of this work). Second, comparing to Zhu’s original design, to integrate the capacitor and resistor on silicon chip was hardly some scientific breakthrough. Engineeringly, it was trivial as long as you have sufficient funding.”

[Response]

We greatly appreciate Reviewer #1’s recognition of the prior work we cited, and we agree that these references offer valuable context for our study. However, we respectfully disagree on his/her assessment of the novelty and significance of our work.

To address the first point, we would like to remark that our work for the first time thoroughly examines the promise of Si-based dual-gate PDs for large-scale in-sensor visual processing of both temporal and spatial visual information at the array level. The novelty (and significance) of our work is not on the structural optimizations at the device (PD) level, but on laying the foundation of large-scale integration of these PDs to form compact, scalable, and CMOS-friendly arrays that can offer in-sensor analog vision processing with high parallelism. In particular, we demonstrate how these arrays of Si-based PDs enable parallelized analog in-sensor processing of both spatial (edge detection) and temporal (event detection) visual information – a capability not yet reported in prior works. Our key contribution in this work is to examine/resolve technical hurdles for array

integration, such as uniformity control, crosstalk suppression, and array-level architecture design. Ample experimental/numerical demonstrations shown in this work suggest the promise of these arrays towards high-throughput computer vision with great impact in retinomorph computing and intelligence sensing. For this reason, while Nat. Electron. 2022 (H. Jang et al) is undoubtedly the pioneering work to report Si-based dual-gate PDs (in contrast to prior dual-gate PDs made of low-dimensional materials), we respectfully think that their study conducted with single kernels does not dampen the novelty of our work on demonstrating parallelized visual processing in both CU arrays and kernel arrays.

To address the second point, we would like to remark that the event-driven pixels reported in Nat. Electron. 2023 (Y. Zhou et al) were built from low-dimensional materials, while being undoubtedly a pioneering work for in-sensor event-sensing, is in contrast to our all-Si based technology that lends itself to CMOS integration, mass production, and large-scale array operation with low variabilities. For this reason, we respectfully think that it is essential to demonstrate the feasibility of achieving compact pixel layout with high filling factors, in which case our effort on monolithic RC integration will ultimately prove beneficial in miniaturized retinomorph systems. We also respectfully remark that efforts on CMOS-compatible array-level integration historically led to significant scientific breakthrough, such as ubiquitous scientific images captured by CMOS-based PD arrays and chip-level genome sequencing enabled by MOSFET-based pH sensor arrays.

To highlight the broad impact of our work, we have added ample experimental and numerical results in this revision (e.g. Figs. 4, 6, and R1, Supplementary Figs. 4 and 19) to exemplify that the analog in-sensor computing capacity of PD-based arrays is well suited to process sophisticated visual objects with high parallelism and well perform the classification tasks among dynamic motions and static images. Such array-level of in-sensor visual processing may shed light on smart sensing systems aimed at large-scale, data-intensive, and latency-sensitive computer vision tasks.

Accordingly, to pinpoint the technical gap of prior works that our technology aims to fill, we have added the following in **Introduction**: “Nonetheless, most of these device prototypes have yet to be tested for both static and dynamic visual processing; their performance in recognizing sophisticated objects at an array level needs to be further examined³³...”. To showcase the promise of our Si-PD based arrays towards visual processing of both dynamic and static objects, we have added two hardware-aware case studies (**new Figs. 4 and 6**) in **1) Results - Promise of large-scale CU arrays for classification of human motions** “Leveraging array-level performance and optimization steps discussed above...thereby holding promise to recognize temporal visual information in real-world settings.”; and **2) Results - Promise of kernel arrays for classification of handwritten digits** “Encouraged by the success of parallelized edge detection at array levels thereby holding promise to recognize spatial visual information with low hardware and power budget.” Taking one step further, we have added a **new Supplementary Fig. 19** to summarize the R_{ph} values of each PD used in these case studies. We hope that these efforts meaningfully address Reviewer #1’s concern on the novelty.

[Reviewer #1, Specific Comment #2]

“(2) There is a severe gap between results presented in Fig.2,3 and Fig.4. Or the logic from Fig.2,3 to Fig.4 is somehow incoherent.

Fig.2 and Fig.3 showed silicon-based DVS cell and small array and the associated measurements. Therefore, from Fig.2 and Fig.3 of this work, as a reader I expected very much that the authors was going to show large-scale array of in-sensor event-detecting. Yet, in Fig.4 the array becomes the ordinary optoelectronic sensors, i.e., light intensity-sensitive, rather than light change-sensitive. And here with such array the authors showed the design of convolutional kernels for edge detection. In my opinion, edge detection was some “change” in spatial continuity, while event-driven refers to change in temporal continuity. From Fig.3e, I understand that some crosstalk effect may hamper the large-scale realization of event-sensing cells. That may explain why in Fig.4 the authors abandoned discussion on the event-sensing and turned back to ordinary light-intensity sensing.

I see that the authors try to argue the parallel computing potential of the array shown in Fig.4. But it is somehow weird to turn from DVS in Fig.2,3 back to ordinary optoelectronic sensor in Fig4. If DVS shown in Fig.2,3 does not apply to large-scale integration, why do authors spend so much discussing it since there was nothing quite new about it?

Thus, is it possible that the authors address the crosstalk effect seriously and therefore realize the large-scale integration circuit of event-sensing cells? I feel that such engineering issue may be the real challenge on the road of neuromorphic camera and addressing it may be the real highlight.”

[Response]

We thank Reviewer #1 for this critical comment on the flow of the paper and the crosstalk issue. To address the first point, we would like to first emphasize that our work aims to present two PD-based arrays for parallelized in-sensor processing of temporal and spatial visual information, respectively. Achieving both using the same device structure and the same material (Si-based dual-gate PDs) could in the longer term prove beneficial for building multifunctional visual processing systems. On the other hand, we respectfully think that edge detection is a non-trivial capability that a neuromorphic imaging system should have, since it can extract spatial information of the objects (complementary to temporal information obtained by event detection) and cut the compute overhead and increase the weight of edge profiles needed to classify the visual objects. It thus makes sense to include both event- and edge-detection demos in our work.

Nonetheless, we do agree with the Reviewer that leaving Figs. 2 and 3 without demonstration on large-scale event detection could lose readers’ interest. Therefore, we have included a **new Fig. 4** to numerically demonstrate the promise of large-scale CU arrays towards precise recognition of human motions in sophisticated environments, with their performance outperforming digital processing methods that map frame differences into binary values through thresholding. To highlight the promise of edge detection (in parallel to event detection), we have also added a **new Fig. 6** to numerically demonstrate the promise of large-scale kernel arrays for precise recognition of handwritten digits, whose performance (94.5 % accuracy) is on par with a traditional CNN-based method.

To address the second point, we would like to emphasize that the crosstalk issue in Fig. 3 can be effectively mitigated by two different approaches included in this appeal, and therefore would not hamper the realization of large-scale CU arrays. In particular, it is possible to reduce this crosstalk by adjusting the gating conditions and/or array structures. In the former, we find that reversing the V_p value in U21 while keeping V_p values in U11 will reduce the crosstalk by *c.a.* 90%

(Supplementary Fig. 17) at the expense of lower temporal resolution in array operation. In the latter, we can build m number of m -by-1 arrays placed in parallel, where CUs from different rows are physically separated from each other to eliminate the crosstalk (Supplementary Fig. 18). New Fig. 4 shows that large-scale CU arrays with these optimization steps are able to recognize different types of human motions (90% accuracy), showcasing their promise to process temporal visual information in real-world settings. We hope that these revisions meaningfully address Reviewer #1's concern on the crosstalk.

Accordingly, we have added: 1) new Figs. 4 and 6 in **Results - Promise of large-scale CU arrays for classification of human motions and Promise of kernel arrays for classification of handwritten digits**, respectively; and 2) two new LTspice simulations as new **Supplementary Figs. 17 and 18** to show that it is feasible to reduce the crosstalk by adjusting the gating conditions and the array structures, respectively. To justify the inclusion of edge detection in the paper, we have also added the following in **Introduction** “Moreover, their optical responses in the analog domain hold promise to capture temporal and spatial visual information needed for visual processing. Nonetheless, most of these device prototypes have yet to be tested for both static and dynamic visual processing...”.

[Reviewer #1, Specific Comment #3]

“(3) what about the algorithm-level innovation adapted to the presented analog DVS?”

In Zhu's original design of differential pair-based DVS, there was no positive/negative threshold component (stepwise functions). What they obtained was hence some kind of analog DVS rather than the digital (or to say polarity) DVS commercially available now. Personally, I feel it is hard to say such analog DVS is advantageous or disadvantageous over the digital DVS. It may depend on the algorithm-level adaptation to the analog output values of the former.

Then, from Fig.2a it seems that the authors of this work totally adopted the design by Zhu and moreover, they outlined “analog” in the paper title. Since analog DVS is fundamentally different from the commercial DVS, could the authors provide some convincing algorithm design to justify that the “analog” circuit proposed in Fig.3 would be of some significant advantages over the digital counterpart by the commercial DVS?*

**Maybe I made a mistake. Zhu et al published the Nat. Electr. work very recently, while the authors' chip should be fabricated before this timing.”*

[Response]

We thank Reviewer #1 for commenting on hardware-aware algorithmic innovation. While our work primarily focuses on advancing hardware for multifunctional analog in-sensor processing, we do recognize that algorithm-hardware co-design can be a powerful approach to maximize the performance of event-based vision sensors. The analog in-sensor processing capability of our PD-based CU arrays lends themselves to rapid, parallelized detection of event-driven light changes with low hardware/power budget. Such array level of low-power visual processing well aligns the requirement of energy- and latency-sensitive neuromorphic algorithms such like SNNs. For this reason, we have tailor designed a hardware-aware SNN algorithm that makes full use of the high

parallelism and bipolar analog output of our CU arrays (see below), and demonstrated its capability in classifying human motions with 90% accuracy. Moving forward, it may be beneficial to achieve fully spatiotemporal processing of the visual objects (e.g., edge- and event-detection to both track and classify unknown objects), which could open new research avenues on software-hardware co-design. To this end, we respectfully think that the CU and kernel arrays presented in our work lay the groundwork for such future studies.

We also thank Reviewer #1 for commenting on if the analog in-sensor visual processing (with bipolar analog readout) in our work is advantageous over its digital counterpart used by the commercial DVS (with unipolar binary readout). To address this comment, we have conducted a numerical study in **new Fig. 4** on examining the promise of our CU-based analog method in detecting and classifying human motions and benchmark the performance with digital DVS methods. Specifically, we format grayscale video clips from the KTH Action dataset (120×160 pixels, 50 frames) based on 25 human subjects making 4 motions (walking, boxing, hand waving, and hand clapping) in a variety of scales and lightning conditions. Next, we randomly select 80 % of these videos to form a training set (the rest forms a validation set), each of which is fed into 10 parallel 120-by-160 CU arrays for analog visual processing. The output of these arrays is emulated by frame differencing, leading ten 120×160 matrices that contain both positive and negative values (*i.e.*, the event detected by each CU). We separately sum positive and negative values in each matrix to form a 20-node input layer of a SNN, whose 4-node output is fed into leaky integrate-and-fire (LIF) neurons to analyze the motion in each frame and finally decide the classification result of the video. The supervised training process of our SNN offers the weight of each CU in 10 arrays, which can be achieved by setting the R_{ph} values of each PD via gating (Supplementary Fig. 19).

Importantly, our SNN trained by bipolar analog output of CUs is able to classify motions in all validation videos with 90 % accuracy, outperforming those trained by digital processing methods that map frame differences into binary values through thresholding (77.5-85 % in new Fig. 4b, using SNNs trained by bipolar digital readout of +1, 0, and -1 to detect both positive and negative light changes). This comparison clearly highlights the advantage of CU-based parallelized analog in-sensor computing in preserving the sub-threshold details that could be lost in digital processing approaches, thereby holding promise to recognize temporal visual information in real-world settings. Furthermore, our analog processing approach fundamentally preserves and process the full range of event-triggered ΔP_{light} , reducing the hardware overhead spent on comparators/counters commonly employed in digital methods. For these reasons, we respectfully think that analog visual processing method could outperform its digital counterpart in dynamic vision tasks with subthreshold details that are essential for object classification and design constraints on hardware/power budget (e.g. resource-limited Internet-of-things systems).

Last but not least, we agree with the Reviewer that our PD-based arrays were independently conceived and demonstrated prior to the publication of Nat. Electron. 2023 (Y. Zhou et al).

Accordingly, to reflect above-mentioned discussions, we have added the following in **Abstract**: “Furthermore, their bipolar analog output captures the amplitude of event-driven light changes and the optical power density on object edges at the device level, a feature that helps boost their performance in classifying dynamic motions and static images.”, and in **Introduction** “Such analog in-sensor computing capability empowers these arrays to process sophisticated visual objects with parallelism and well perform the classification tasks among dynamic motions and static images... We numerically show that these CUs can form ten 120-by-160 arrays to classify

sophisticated human motions (with 90 % accuracy) via an offline spike neural network (SNN).”.
To showcase the promise of our PD-based arrays towards visual processing of both dynamic and static objects, we have added two hardware-aware numerical case studies (new Figs. 4 and 6) in Results - Promise of large-scale CU arrays for classification of human motions and Promise of kernel arrays for classification of handwritten digits, respectively, and furthermore added technical details in Methods - Simulations on the CU array for classifying dynamic motions and Simulations on the kernel array for classifying static images.

B. RESPONSE TO REVIEWER #2

[Reviewer #2, General Comment]

“In this manuscript, the authors introduce two scalable in-sensor visual processing arrays based on dual-gate amorphous-silicon photodiodes for multiplexed event sensing and edge detection of visual objects. These arrays can process both temporal and spatial visual information in an analog manner, mimicking the functionality of the human retina. The concept of in-sensor computing presents a significant advancement for next-generation computer vision hardware, and this paper makes an innovative contribution to this field. The paper provides detailed technical information in the Methods and Supplementary Information sections, enabling readers to reproduce the research. Overall, I believe this manuscript is well-prepared and organized. I would like to recommend this manuscript for publication in Nature Communications. Here are some specific comments and suggestions:”

[Response]

We thank Reviewer #2 for acknowledging the significance, novelty, and merit of our work.

[Reviewer #2, Specific Comment #1]

“Q1. I am curious about why two different wavelengths (530 nm, 595 nm) of light were used in Fig. 3. Based on my understanding of this article, two superimposed light pulses with the same wavelength can also produce the same result.”

[Response]

We agree with the Reviewer that site-specific event sensing in Fig. 3 can also be achieved by two light sources with the same wavelength. The reason we choose two different wavelengths here though is twofold: 1) since α -Si based PDs have a broad absorption spectrum in the visible regime, we hope to examine if our CU arrays can respond to event-driven light changes at different wavelengths, which are often the case in real-world settings (e.g. bioimaging, automated navigation); 2) choosing different wavelengths helps confirm the 530-nm light spot (local event) is spatially confined on U11 only (not on other CUs).

[Reviewer #2, Specific Comment #2]

“Q2. Please review 'at 550/15 nm' on page 13, as it confused me. Is it a typo?.”

[Response]

We thank Reviewer #2 for this remark. '550/15 nm' on Page 13 is the notation commonly used in scientific imaging, which refers to the illumination light being centered at 550 nm with a 15 nm bandwidth. The 550/15 nm illumination in our work is generated by a SPECTRA X light engine with a bandpass filter.

Accordingly, we have added the following in **Results - optoelectronic characteristics of as-made PDs**: “Next, under constant optical power density ($P_{\text{light}} = 530 \text{ mW/cm}^2$ at 550/15 nm, an illumination centered at 550 nm with a 15 nm bandwidth)...”.

[Reviewer #2, Specific Comment #3]

“Q3. I suggest that the authors redraw Fig. 4d and 4g to make them more easily understandable.”

[Response]

We thank Reviewer #2 for this great suggestion. Accordingly, we have redraw previous Figs. 4d and 4g as the new Figs. 5e and 5f with two major improvements: 1) clearly mark the positions of 64 kernels and the light spots used in the experiments; 2) clearly map out the polarities of R_{ph} values in each kernel and project them to the maps of $V_{\text{H/V}}$. We hope these efforts will make the scientific contents more accessible to readers.

[Reviewer #2, Specific Comment #4]

“Q4. Can the uniformity of the PD devices or the variations in the fabrication process be well-controlled? Since the multiple PD cells work collaboratively, it is important for their performance to be similar to each other.”

[Response]

We thank Reviewer #2 for commenting on the uniformity of PD devices. Since our Si-based PDs are fabricated by CMOS-compatible processes that are established for wafer-scale production of semiconductor devices, we would expect their uniformity/variability of the performance suffices the need for large-scale array operation. Moreover, we have adopted an error-tolerant PD layout by placing an even number of α -Si channels between each pair of the S- and D-contacts, which also serves to mitigate device-to-device variabilities (Fig. S2). Consequently, we observe 7-9 % variations of PD readout at $V_p = -3 \text{ V}$ and 3 V across 36 PDs from 4 kernels (Supplementary Fig. 23), suggesting a high uniformity on par with the wafer-level statistics reported in Nat. Electron. 2022 (H. Jang et al, 10-13 %). Taking a step further, we observe that the gate-tunability of our PDs offers a powerful approach to cancel the variability effect at the CU/kernel level (Supplementary Tabs. I, II). Specifically, by adjusting V_p values applied to each PD, we can readily achieve: 1) the same amplitude of photocurrents in two branches of the CU; and 2) the expected 9 R_{ph} values in each kernel for convolutional filtering. For these reasons, we respectfully think that the variability of our PDs can be well controlled and furthermore canceled out by gate tuning, hence not forming an issue for array operations. Moving forward, the variability effect in PD-based arrays could be further studied by: 3) designing compensation circuits to cancel the DC offset in CU readout; or 4) optimizing the gate control at the array level to finetune the responses

of each kernel. We hope that these discussions and experimental evidence meaningfully address Reviewer #2's concern on the uniformity of PDs.

Accordingly, we have added the following in Discussion: "The uniformity of PDs in our array is on par with prior wafer-level statistics²⁹(Supplementary Fig. 23), and can further be controlled via gate tuning (Supplementary Tabs. I, II)."

[Reviewer #2, Specific Comment #5]

"Q5. The zero-bias condition ($V_S = V_D = 0 V$) is very promising for low-power edge applications. I am curious about the possibility of achieving a zero-power intelligent in-sensor computing system. Could the authors provide a discussion or outlook on this point?"

[Response]

We thank Reviewer #2 for highlighting the promise of our technology towards low-power edge applications. Our in-sensor visual processing arrays based on zero-biased computing units, if co-designed with low-power readout circuits in future studies, could indeed present a viable approach to build low-power edge systems (e.g., networked mobile platforms) that are capable of intelligent computing. Accordingly, we have added the following in Discussion: "Moving forward, our analog in-sensor visual processor arrays can add to the advancement of to build low-power edge systems (e.g., mobile platforms) for intelligent computing".

C. RESPONSE TO REVIEWER #3

[Reviewer #3, General Comment]

"In this manuscript, the authors report on in-sensor visual processing arrays of dual-gate amorphous-silicon photodiodes, used for multiplexed event sensing at sub-ms precision and edge detection of multiple objects, respectively. The device arrays are capable of processing both temporal and spatial visual information. It is an important topic in the area of vision systems, requiring improved vision systems with reduced latency and improved performance. Some valuable engineering work can be seen in integrating the PD and achieving different spiking polarities in an appropriate circuit form, but it incurs area penalty that novel devices can achieve in a single device. As such, technological and scientific advancement might seem to be limited in the current version. My comments to the authors are as follows:"

[Response]

We thank Reviewer #3 for highlighting the significance and merit of our work, and would like to delineate/strengthen the technological and scientific advancement of this work through this revision. We will also address the area penalty in the response to Reviewer #3's Specific Comment #5 and #7 (see below).

[Reviewer #3, Specific Comment #1]

"1. It is mentioned that a-Si is chosen for its high absorption coefficient and its ability for 3D stacking, which is important for high-density integration. However, the use of a-Si brings about problems, including defects, non-uniformity, performance, etc. Can the authors comment?"

[Response]

We thank Reviewer #3 for this insightful remark. We agree with the Reviewer that the use of α -Si could be suboptimal due to its defects and the resulting non-uniformity, etc. Nonetheless, as mentioned in our response to Reviewer #2's Comment #4, we respectfully think that the uniformity of α -Si based PDs can be well controlled via CMOS-compatible fabrication steps and gate tuning, hence not forming an issue for array operations.

Moving forward though, it may be interesting to crystallize α -Si into polycrystalline Si via laser annealing steps (a widely adopted low-temperature process for CMOS compatibility^{A1-A4}), which might help circumvent the issues of defects and non-uniformity among α -Si films.

[Reviewer #3, Specific Comment #2]

"2. Can authors explain the reason for using 595 nm light in their experiment? What would the different wavelengths (visible light) have on the behavior of the PD I-V?"

[Response]

We thank Reviewer #3 for commenting the choice of wavelengths. Since α -Si based PDs have a broad absorption spectrum in the visible regime, we select 595 nm in Figs. 1 and 3 to merely for illustrative purposes (i.e., to exemplify the PD response in the visible spectrum; 530 nm and 550 nm are also used in our work). Choosing other visible wavelengths are expected to yield a qualitatively similar PD I-V, while the I_{ph} values may proportionally change with the absorption coefficients of α -Si at those wavelengths.

Experimentally, we have collected I_S-V_p curves of a PD under 440/20 nm (522 mW/cm²), 470/24 nm (400 mW/cm²), 550/15 nm (530 mW/cm²), and 640/30 nm (471 mW/cm²) and plotted the third sweep in Fig. R1 (see below), which showcase such wavelength-dependent PD I-Vs.

Figure R1. $I_S - V_p$ curves of a single PD under different wavelengths

[Reviewer #3, Specific Comment #3]

"3. In Figure 1b, the energy band diagram of the PD is drawn. However, the gate biases applied change the regions to both n and p type. If the device is not under thermal equilibrium conditions, why would the fermi level line up?"

[Response]

We thank Reviewer #3 for this remark. The Fermi level of the PD lines up because we bias its S- and D-contacts with $V_S = V_D$ (i.e., short-circuited). The band structure of a short-circuited dual-

gate PD should be equivalent to that of a zero-biased chemically-doped PIN diode, where photon-induced excess carriers (i.e. electron-hole pairs) are separated by the built-in field across the PD.

[Reviewer #3, Specific Comment #4]

“4. In supplementary figure 4, the same bias is applied to both gates and I-V obtained and illuminated. Have the authors investigated the situation where no light is illuminated?”

[Response]

We thank Reviewer #3 for this remark. Accordingly, we have measured I-Vs in the dark (Fig. R2, see below) and add them to Supplementary Figure 4. As expected, the measured dark current is orders smaller than I_S measured under 550/15 nm illumination; the dark current measured at $V_{G1} = V_{G2} = 3$ or -3 V are higher than those measured with floated V_{G1} and V_{G2} due to the similar reasons discussed in the text. Interestingly, we find that the dark current nonlinearly increases with the amplitude of V_P , in contrast with I_S measured under 550/15 nm illumination. This result suggests the existence of nontrivial Schottky barriers near the S- and D-contacts (i.e., Ti/Pt-Si junctions) in the dark, which can be effectively lowered by the applied 550/15 nm illumination^{A5}.

Figure R2. $I_S - V_S$ curves of a single PD when $V_{G1} = V_{G2} = -3$ or 3 V and that when both G1- and G2-contacts are floated measured in the dark.

[Reviewer #3, Specific Comment #5]

“5. In figure 2a, the spikes are generated with the aid of passive elements and TIA, this makes a cell somewhat bulky. Such behavior has also been in some reported works on 2D materials where the events can be detected at different light intensities.”

[Response]

We thank Reviewer #3 for this critical comment on the area penalty and prior works based on 2D materials.

For the area penalty caused by passive RCs, we note that the areas they take could be reduced by adjusting their values and layout. For instance, we have numerically shown (Supplementary Fig. 8) that a sub- μ s resolution of event detection can be achieved by CU arrays with smaller R_s (by cutting their lengths) and C_s (by cutting their areas), both of which can effectively reduce the areas these RCs take. On the other hand, we can also cut the areas of each CU by stacking PDs on top of RCs, which would effectively increase FF values^{A6-A8}. For the area penalty caused by TIAs, we note that they do not need to be monolithically integrated into each CU; instead we can in the future allow each TIA to be shared by one row of CUs (1-by- m , m is the number of CUs per row) at the expense of lowering the temporal resolution. In sum, we agree with the Reviewer that the

compactness of CUs in Fig. 2a is suboptimal (chosen to showcase the CU function only), and hope that these discussions can meaningfully address the concern on area penalty.

We also agree with the Reviewer that spiking-based event detection has been achieved by pixels built from 2D materials, such as those reported in Nat. Electron. 2023 (Y. Zhou et al). However, we remark that our work for the first time thoroughly examines the promise of Si-based PDs for large-scale in-sensor visual processing of both temporal and spatial visual information at the array level (see Reviewer #1's Comment #1). Notably, our Si-based technology lends itself to the formation of compact, scalable, and CMOS-friendly arrays that can offer in-sensor analog vision processing with high parallelism. For this reason, we respectfully think that prior works on 2D-material based devices (e.g., Nat. Electron. 2023) do not dampen the impact of our work on establishing all-Si based array technologies towards high-throughput computer vision.

Accordingly, we have added the following in the **Discussion** to address area penalty: “Fourth, the compactness of our CU array can be further improved by stacking PDs on top of the RC elements, choosing smaller Rs and Cs⁴⁰, integrating on-chip TIAs for each row of CUs, or using multilayer metallization to reduce the areas occupied by gate lines.”. To highlight the technical gap of prior works (including those on 2D materials) that our technology aims to fill, we have also adding the following in the **Introduction** section: “Nonetheless, most of these device prototypes have yet to be tested for both static and dynamic visual processing; their performance in recognizing sophisticated objects at an array level needs to be further examined³³. For these reasons, it is imperative to develop ideally scalable, compact, and low-power arrays to detect dynamic events and extract static features with a high degree of parallelism. This is a non-trivial task as it requires a holistic modular-array co-design that take key figure-of-merits (e.g., uniformity, crosstalk, power) into a well-balanced account.”

[Reviewer #3, Specific Comment #6]

“6. In figure 2, the error bar associated to the performance of the PD appears to be quite large, especially in Figure 2c (for example) between the on spike and off spikes. Can the authors comment on the reliability of using such device in large arrays.”

[Response]

We thank Reviewer #3 for commenting on the error bars and the reliability of using our PDs for large arrays. To address this critical comment, we have conducted additional circuit simulations using the setup in Supplementary Fig. 9 and examined the CU readout traces we experimentally collected in Fig. 2. Our results show that: 1) among three 20-pulse periods, $|A_{\text{on}}|/|A_{\text{off}}|$ values of the 1st ON/OFF spike in each period are higher than those of the rest 19 spikes. This result is likely due to incomplete charging/discharging processes of the capacitor C between neighboring light pulses, which become less severe in CUs with smaller RC values (Fig. R3). Therefore, we could choose smaller RC values in future studies to improve the reliability of $|A_{\text{on}}|$ and $|A_{\text{off}}|$ (with smaller errors), which can also serve to achieve higher temporal resolutions and reduce the heat dissipation (see Discussion); 2) the difference between $|A_{\text{on}}|$ and $|A_{\text{off}}|$ (Figs. 2c-d) is likely related to the TIA gain (i.e., transimpedance) we chose to trace the CU readout, since lowering the gain is found to effectively reduces their difference (Fig. R4). This is possibly because the amount of discharging current across the capacitor C that flows into the TIA starts to increase (and thereby add to $|A_{\text{off}}|$) when the TIA transimpedance gets lower than the impedance of the 1R1C branch. In practice, we

can lower the TIA gain in future studies (according to select RC values) to reduce the mismatch between $|A_{\text{on}}|$ and $|A_{\text{off}}|$; 3) the error bars of $t_{\text{rise/fall}}$ are largely due to limited signal-to-noise ratios (SNR) in CU traces, especially those measured at low ΔP_{light} and low V_p values (Fig. R5); the high-bandwidth mode of our TIA (SR570) also introduced high-frequency noises during experiments. For this reason, we change to plot the average of t_{rise} and t_{fall} in Figs. 2c-d quantify the temporal resolution of our CUs, which appears to partially cancel the errors in the values of t_{rise} or t_{fall} alone and result in smaller error bars.

On the other hand, the reliability of PDs for array operation also relies on their uniformity /variability across the array. To this end, we respectfully think that the variability among our PDs can be well controlled and furthermore canceled out by gate tuning, hence not forming an issue for array operations (see detailed response to Reviewer #2's Comment #4). Moving forward, the variability effect in PD-based arrays could be further minimized by: 1) designing compensation circuits to cancel the DC offset in CU readout; or 2) optimizing the gate control at the array level to finetune the responses of each kernel. We hope that these discussions, experimental/numerical evidence, and analysis meaningfully address Reviewer #3's concern on the reliability of PDs.

Figure R3. RC values are found to affect the reliability of $|A_{\text{on}}|$ and $|A_{\text{off}}|$ in V_{out} traces of a single CU. PDs are modeled the same way as Supplementary Fig. 9b with $R_p = 100 \text{ G}\Omega$, $R_s = 100 \text{ k}\Omega$, and $C_j = 12 \text{ pF}$.

Figure R4. TIA gain is found to affect the mismatch between $|A_{\text{on}}|$ and $|A_{\text{off}}|$ in V_{out} traces of a single CU. PDs are modeled the same way as Fig. R3.

Figure R5. Representative V_{out} traces of the CU tested in Fig. 2, whose SNRs vary with V_p and ΔP_{light} values.

Accordingly, we have added the following in **Results - Pairing dual-gate PDs for analog in-sensor event detection**: “The mismatch between $|A_{on}|$ and $|A_{off}|$, while not affecting event sensing, can be effectively reduced by optimizing the experimental setting (Supplementary Fig. 8)”, and added Figure R4 as the **new Supplementary Fig. 8**. Additionally, we have added the following in **Results - Pairing dual-gate PDs for analog in-sensor event detection**: “The variability among time constants suffices the demonstration purpose here, but can be reduced by optimizing the testing conditions”, and added Figure R5 as the **new Supplementary Fig. 10**.

[Reviewer #3, Specific Comment #7]

“7. It is noted that the gates in the arrays are independently addressed; would this be a limiting factor for large arrays?”

[Response]

We thank Reviewer #3 for commenting on the gating strategy. We respectfully think that there is tradeoff between the flexibility in programming individual CUs/kernels and the area penalty their gate lines may occupy in large arrays. The former is equally important since the gate tunability of individual CUs/kernels are critical features to cancel out uniformity/variability at the array level (see Reviewer #2’s Comment #4). Moreover, we respectfully think that the area penalty from gate lines can be less of an issue if we: 1) fabricate them using the multilayer metallization strategies adopted in CMOS processes; or 2) partially common gate lines among local groups of CUs/kernels since their variability can be well controlled via CMOS-friendly processes (e.g., common gate lines in the 1R branch and the 1R1C branches of local CUs, respectively). We hope that these discussions meaningfully address Reviewer #3’s concern on the gating strategy. Accordingly, we have added the following in the **Discussion** to address area penalty: “Fourth, the compactness of our CU array can be further improved by stacking PDs on top of the RC elements..., or using multilayer metallization to reduce the areas occupied by gate lines.”.

[Reviewer #3, Specific Comment #8]

“8. With the current array design, can the crosstalk be eliminated with appropriate gate biases? This has not been discussed in the manuscript.”

[Response]

We thank Reviewer #3 for this insightful advice on reducing the crosstalk. Accordingly, we have conducted additional circuit simulations using the setup depicted in Supplementary Figs. 16d, and found that the crosstalk discussed in Fig. 3 can be largely eliminated by reversing the polarity of V_p values in the non-targeted CU (Fig. R6). We respectfully think that this approach suggested by Reviewer #3 and our previous idea of replacing the cross-barred CU array by multiple 1-by- m CU arrays placed in parallel can both effectively mitigate the crosstalk issue (see circuit diagrams in Fig. R7). Accordingly, we have added the following in **1) Results - Parallel event detection with in-sensor CU arrays**: “Nonetheless, this crosstalk can be mitigated by adjusting the gating conditions and/or array structures. In the former, we find that reversing the polarity of V_p in U21 while keeping V_p values in U11 will reduce the crosstalk by *ca.* 90%”, and added Fig. R6 as the **new Supplementary Fig. 17**. We have also added the following in **2) Results - Parallel event detection with in-sensor CU arrays**: “In the latter, we can place 1-by- m arrays in parallel with m

being the number of columns, where CUs from different rows are physically separated from each other to eliminate the crosstalk”, and included Fig. R7 as the **new Supplementary Fig. 18**.

Figure R6. LTspice simulation of the crosstalk mitigation in Supplementary Fig. 16d. PDs in U1 and U2 are modeled the same way as Supplementary Fig. 9. In the red curve, PDs in U1 and U2 are biased at $V_{G1} = -V_{G2} = 2.5$ V and a 1 nA current pulse ($t_{on/off} = 50/50$ ms) switching between its 10 and 90 % amplitude within 1 ns. In the yellow curve, PDs in U1 are biased the same as before but those in U2 are biased at $V_{G1} = -V_{G2} = -1.4$ V and a -0.56 nA current pulse ($t_{on/off} = 50/50$ ms) switching between its 10 and 90 % amplitude within 1 ns (note: the current value matches those measured from our fabricated PDs). Our data show that the crosstalk at non-targeted U2 (with light illumination on U1 only) can be significantly reduced if we reverse the polarity of its gate biases.

Figure R7. LTspice simulation of the crosstalk from a 1-by-8 CU array. PDs in U1-U8 are modeled the same way as Supplementary Fig. 9.

[Reviewer #3, Specific Comment #9]

“9. Especially for matured materials made devices, I think that a proper benchmark with reported state of art is important, highlighting the advantages of current work. A lot has been claimed in the discussion section on scalable in-vision processor array and functions. These have to be

properly compared as well or the claim might appear to be weak.”

[Response]

We thank Reviewer #3 for this great advice and have added 2 benchmark tables to Supplementary Information (Supplementary Tabs. III and IV), focusing on the filling factors in the state-of-the-art CMOS-imager based edge- and event-detection technologies and our works. In sum, our Si-PD based array technology can be noted for its zero-bias operation, high FFs (30% for CUs and 90% for kernels), and the ability to process both temporal and spatial information in the analog domain. Moving forward though, our technology will certainly benefit from further optimizations on its electrical crosstalk, uniformity control, gating strategy, area penalty, and material choices. In addition, we have cited these tables in Discussion: “We have presented two scalable in-sensor visual processor arrays based on a compact modular design (Supplementary Tabs. III and IV) of α -Si based dual-gate PDs for parallelized analog processing of temporal and spatial visual information (i.e. events and edges), respectively”.

[Reviewer #3, Specific Comment #10]

“10. The authors should demonstrate more applications regarding static image processing and event-based motion processing based on this array.”

[Response]

We thank Reviewer #3 for making this very helpful suggestion to further strengthen this work. Accordingly, we have added two hardware-aware numerical case studies in this revision (new Figs. 4, 6) to exemplify that the analog in-sensor computing capacity of our PD-based arrays is able to not only process sophisticated visual objects with high parallelism, but also outperform classification tasks among dynamic motions and static images.

In the former (new Fig. 4), we find that a SNN trained by bipolar analog output of CUs is able to classify motions in 2-s videos with 90 % accuracy, outperforming those trained by digital processing methods that map frame differences into binary values through thresholding. This result highlights the advantage of CU-based parallelized analog in-sensor computing in preserving the sub-threshold details that can be lost in digital processing approaches, thereby holding promise to recognize temporal visual information in real-world settings.

In the latter (new Fig. 6), we find that an ANN trained by bipolar analog output of the kernel array can classify handwritten digits with 94.8 % accuracy, on par with a CNN based on scanning single kernels across the images and outperforming a NN trained by original. This result suggests the advantage of kernel-based parallelized analog in-sensor computing in cutting the compute overhead (vs. CNN) and increasing the weights of key spatial details in the image (vs. original images), thereby holding promise to recognize spatial visual information with low hardware and power budget.

Accordingly, we have added the following in **1) Results - Promise of large-scale CU arrays for classification of human motions:** “Leveraging array-level performance and optimization steps discussed above ... thereby holding promise to recognize temporal visual information in real-world settings.”; and **2) Results - Promise of kernel arrays for classification of handwritten digits:** “Encouraged by the success of parallelized edge detection at array levels thereby holding promise to recognize spatial visual information with low hardware and power budget.”

D. INCORPORATION OF EDITOR'S COMMENTS

[Editor Comment #1]

"...In this regard, although the reviewers find your work to be of some interest, all of them raise critical concerns on the design and its broad implication for applications. In particular, you will see that both Reviewers #1 and #3 cast serious doubt on the effect of crosstalk on realizing large-scale event-sensing arrays. Meanwhile, Reviewers #2 and #3 questions the uniformity of the device. Indeed, it seems that a significant amount of additional experimental work and clarification will be necessary in order to support your key claims.."

[Response]

We have thoroughly followed and addressed all reviewers' remarks, comments, and suggestions with ample new experimental/numerical data and critical discussions as we have detailed above.

In particular, we have highlighted the broad impact of our work (Reviewer#1 Comment#1), proposed two complementary methods to mitigate the crosstalk issue (Reviewer#1 Comment#2, Reviewer#3 Comment#8), and delineated the uniformity/variability/reliability issue by new data in the Supplementary Information (Reviewer#2 Comment#4, Reviewer#3 Comment#6). We hope that these efforts can meaningfully address Editor's concern on this manuscript.

[Editor Comment #2]

"However, we would not rule out the consideration of a fully and thoroughly revised manuscript in future that presents a stronger case on the reliable demonstration of large-scale event sensing with decent uniformity."

[Response]

We thank Editor for offering the opportunity to reconsider our revised manuscript. We have thoroughly incorporated this advice in the revision, as we have detailed above in our response to all three reviewers' comments. In particular, new experimental and numerical evidence suggest the promise of our presented technology towards reliable large-scale array operation with decent uniformity (e.g. via optimized RC values, circuit gains, and gate biases to cancel the variability). We hope that the data, analysis, and clarifications in this revision can present a stronger case and meaningfully address Editor's concern on the original submission.

E. OTHER REVISIONS

- We have thoroughly revised **Abstract, Introduction, Results, Discussion, Methods** of the original manuscript to improve the readability (with no changes in the scientific merits).

Additional References:

- A1. Ramanathan, A. K. *et al.* Monolithic 3D+-IC based massively parallel compute-in-memory macro for accelerating database and machine learning primitives. *2020 IEEE International Electron Devices Meeting (IEDM)*. San Francisco, CA, USA, pp. 28.5.1-28.5.4, IEEE, 2020.
- A2. Hsueh, F. K. *et al.* First demonstration of ultrafast laser annealed monolithic 3D gate-all-around CMOS logic and FeFET memory with near-memory-computing macro. *2020 IEEE International Electron Devices Meeting (IEDM)*. San Francisco, CA, USA, pp. 40.4.1-40.4.4, IEEE, 2020.
- A3. Hsueh, F. K. *et al.* Monolithic 3D SRAM-CIM macro fabricated with BEOL gate-all-around MOSFETs. *2019 IEEE International Electron Devices Meeting (IEDM)*. San Francisco, CA, USA, pp. 3.3.1-3.3.4, IEEE, 2019.
- A4. Chang, S. W. *et al.* First demonstration of CMOS inverter and 6T-SRAM based on GAA CFETs structure for 3D-IC applications. *2019 IEEE International Electron Devices Meeting (IEDM)*. San Francisco, CA, USA, pp. 11.7.1-11.7.4, IEEE, 2019.
- A5. Lu, MY. *et al.* Quantifying the barrier lowering of ZnO Schottky nanodevices under UV light. *Sci Rep*, **5**, 15123 (2015).
- A6. Ibbotson, D. *et al.* Manufacturability optimization and design validation studies for FPGA-based, 3D integrated circuits. *2013 Symposium on VLSI Technology*. Kyoto, Japan, pp. T38-T39. IEEE, 2013.
- A7. Liao, W. S. *et al.* A manufacturable interposer MIM decoupling capacitor with robust thin high-K dielectric for heterogeneous 3D IC CoWoS wafer level system integration. *2014 IEEE International Electron Devices Meeting*. San Francisco, CA, USA, pp. 27.3.1-27.3.4, IEEE, 2014.
- A8. Hou, S. Y. *et al.* Integrated deep trench capacitor in Si interposer for CoWoS heterogeneous integration. *2019 IEEE International Electron Devices Meeting (IEDM)*. San Francisco, CA, USA, pp. 19.5.1-19.5.4, IEEE, 2019.

Point-by-Point Response to Reviewers' Comments

A. RESPONSE TO REVIEWER #1

[Reviewer #1, General Comment]

The authors have addressed all of my concerns and the revisions made by the authors greatly amplify the innovative aspects of their research. I suggest the publication of this work.

[Response]

We thank Reviewer #1 for recommending our revised manuscript for publication.

B. RESPONSE TO REVIEWER #2

[Reviewer #2, General Comment]

I read the response carefully, and found that all my comments have been well addressed in the revised manuscript. I have no further questions and am happy to recommend it for publication.

[Response]

We thank Reviewer #2 for recommending our revised manuscript for publication.

C. RESPONSE TO REVIEWER #3

[Reviewer #3, General Comment]

Comments are generally answered satisfactorily, but I have the following follow up questions.

[Response]

We thank Reviewer #3 for acknowledging the efforts we made in this revision, and would like to address the follow-up comments in the following.

[Reviewer #3, Specific Comment #1]

1. Authors replied on the use of laser annealing to convert amorphous Si to poly Si. I think authors need to think carefully on the reply to this comment as there is a clear difference in behavior (eg. Absorption) for both materials. What kind of material is more suited for the application presented in the work, and not just to mitigate the non-uniformity issues. Hence, this part needs to be better justified in the manuscript.

[Response]

We thank Reviewer #3 for commenting on material properties. However, we respectfully think that poly-Si prepared by laser annealing may in fact hold promise to form CMOS-compatible PDs with broad absorption spectra across visible and near-IR ranges. Comparatively, poly-Si (higher carrier mobilities) and α -Si (higher absorption coefficients) may each lend itself to high-speed and low-light applications, respectively. Importantly, both materials are fully compatible to Si-based

CMOS fabrication steps, and can form photo-sensitive areas of the PDs on top of the routing wires and/or CMOS-based integrated circuits that may otherwise reduce the filling factors.

Nonetheless, we do agree with the Reviewer that other CMOS-compatible materials (e.g. SiGe) may in the future add to integrated retinomorphic arrays for targeted applications. Accordingly, to reflect above-mentioned discussion and leave the manuscript with a focused scope, we have added the following in **Discussion**: “(other CMOS-compatible or low-dimensional materials may also be chosen to build application-specific arrays in the future)”.

[Reviewer #3, Specific Comment #2]

2. I hope authors are aware that 2D materials are scalable as well and added discussion does not seem to capture the differences between Si and 2D materials clearly. Otherwise it will be misleading to readers and hinder the further advancement of this technology.

[Response]

We thank Reviewer #3 for commenting on other materials. We agree with the Reviewer that 2D materials share the same scalability as Si and hold great promise in the field of in-sensor visual processing. However, we also respectfully think that they could benefit from further studies on examining their temporal precision, reliability, and feasibility for mass production of large-scale arrays as we have noted in the manuscript (e.g. uniformity control, crosstalk suppression, and CMOS-integration strategies). Accordingly, to reflect above-mentioned discussion and leave the manuscript with a focused scope, we have added the following in **Discussion**: “(other CMOS-compatible or low-dimensional materials may also be chosen to build application-specific arrays in the future)”.

[Reviewer #3, Specific Comment #3]

3. Also, while it is true that multi-metallization can be used to address individual pixels but you cannot avoid the low Si area efficiency for arrays that needs to be addressed in this manner as the number of I/O required cannot be reduced. I think this disadvantage needs to be mentioned as well.

[Response]

We thank Reviewer #3 for commenting on area penalty. In this regard, we would like to remark that: 1) the FF in our kernel arrays is > 90 % since we leave no metal contacts on top of the photo-sensitive α -Si regions; the FF in our CU arrays (> 30 %) can be further increased by stacking PDs on top of the RC elements, choosing smaller Rs and Cs, or using multilayer metallization to reduce the areas occupied by gate lines (see **Discussion**). Therefore we respectfully think that the Si area efficiency will not hinder the formation of large-scale arrays; 2) gate routing lines and row/column-connecting lines in our arrays can all be routed beneath photo-sensitive α -Si regions (i.e. PDs are built on top of routing wires), therefore not forming an issue to lower the FF values; 3) the number of TIAs required to operate the CU/kernel arrays is no more than those used in the state-of-the-art CMOS imagers (e.g. one TIA shared by one column of PDs), in which case the number of I/Os does not hamper the formation of large-scale arrays. For these reasons, we respectfully think that our strategy of addressing individual pixels will not be disadvantageous for large-scale array operation due to Si area efficiency and/or the number of I/Os.